# Community structure-regulation coupling reveals optimal information diffusion

Xiaojie Chen [1,2,9], Meiling Xie [1,9], Jun Meng[3], Sheng Fang[1], Xiaosong Chen [1,4], Jürgen Kurths [5,6], Jan Nagler [7] ✉ & Jingfang Fan [1,5,8] ✉

Effective regulation of information diffusion in complex social systems requires balancing containment and intervention cost, yet how network community structure interacts with targeted interventions remains unclear. We develop a community structure-regulation coupling framework (COSREF) that integrates community structure with process-level regulation of transmission and show how their interplay governs diffusion. Tuning two regulation parameters governing within- and cross-community transmission yields three regimes: no, localized, and global diffusion, separated by abrupt transitions. This structure–regulation perspective reveals a low-cost intervention region where small, targeted adjustments contain spread, unifies topology and regulation within a single theoretical setting, and provides general principles for efficiently and robustly regulating modular systems. Analyses of large cross-platform real-world social networks confirm our analytical predictions and simulation results, demonstrating COSREF's robustness across investigated topologies and its applicability to real information environments.

The rapid spread of misinformation poses a profound threat to scientific integrity, public health, and even democratic governance worldwide[1,2]. Intervention efforts have often targeted a small subset of "super-spreaders", yet such measures risk provoking public concerns over freedom of speech[3]. Common blocking strategies, such as isolating key influencers, disrupting transmission channels, and promoting factual information, have shown only limited success[4,5]. Moreover, large-scale restrictions incur substantial social, political, and economic costs[6,7]. Balancing effective suppression with minimal societal disruption, therefore remains a central challenge for modern information governance.

Recently, a rich variety of regulation strategies have been proposed, ranging from network blocking and algorithmic downranking to cross-community throttling and AI-based misinformation detection[1,8–16]. Yet despite this diversity, the structural heterogeneity of real social networks remains a fundamental obstacle. The resulting coupling between structure and dynamics[17,18] renders truly optimal regulation strategies[19–21] elusive—even aside from the massive data-fitting challenges inherent to practical implementation. Fundamentally, information diffusion, encompassing the spread of news, rumors, and viral content, is deeply embedded in the topology of social and digital networks. Network science has provided critical insights into how connectivity patterns shape diffusion outcomes[6,22]. In such representations, nodes correspond to individuals or platforms, and links capture social interactions or digital exchanges.

Among network features, community structure plays a decisive role in diffusion processes by shaping the balance between local reinforcement and long-range spreading. Strong community structure

[1]School of Systems Science/Institute of Nonequilibrium Systems, Beijing Normal University, 100875 Beijing, China. [2]School of Physics, Hubei University, 430062 Wuhan, Hubei, China. [3]State Key Laboratory of Earth System Numerical Modeling and Application, Institute of Atmospheric Physics, Chinese Academy of Sciences, Beijing 100029, China. [4]Institute for Advanced Study in Physics, Zhejiang University, Hangzhou 310058, China. [5]Potsdam Institute for Climate Impact Research, 14412 Potsdam, Germany. [6]Department of Physics, Humboldt University, 10099 Berlin, Germany. [7]Deep Dynamics, Centre for Human and Machine Intelligence, Frankfurt School of Finance & Management, 60322 Frankfurt am Main, Germany. [8]State Key Laboratory of Marine Environmental Science, Xiamen University, 361005 Xiamen, China. [9]These authors contributed equally: Xiaojie Chen, Meiling Xie. ✉e-mail: jan.nagler@gmail.com; jingfang@bnu.edu.cn

facilitates local reinforcement and thereby promotes cluster-level adoption, whereas weak community structure enhances long-range spreading through inter-community bridges[23]. Multilayer connectivity[24,25] and bursty temporal dynamics[22] can further reshape these effects by altering patterns of exposure over time. Related phenomena also arise in epidemic models, where interlayer coupling can sustain endemic states even when individual layers remain below the epidemic threshold[26,27], and strongly connected communities can either suppress or accelerate infection depending on the governing dynamics[28].

Recent studies have shown that community structure can fundamentally reshape the dynamics of complex contagion. In particular, Nematzadeh et al.[23] demonstrated that community structure alone can produce nontrivial cascade behaviour, revealing that intermediate levels of community mixing may maximize the probability of global spreading. That work established a foundational framework for understanding uncontrolled diffusion dynamics on modular networks. However, how community structure interacts with explicit regulatory interventions remains largely unexplored.

To address this gap, we develop a general community structure–regulation coupling framework (COSREF) (Fig. 1). As a substantial extension to the aforementioned modular contagion models, we introduce an additional dimension to the problem by incorporating explicit parameters that modulate transmission. Specifically, our framework links (a) community structure, (b) threshold-based contagion dynamics, and (c) two regulatory parameters, $\omega_{intra}$ and $\omega_{inter}$, which govern intra- and inter-community influence, respectively.

Through theoretical analysis and simulations, we investigate how this interplay between network topology and regulatory mechanisms shapes diffusion outcomes. We show that tuning these parameters yields three distinct diffusion regimes—non-diffusive, localized, and global—separated by abrupt transitions. Consequently, beyond the classical cascade phase diagram, our framework reveals the existence of an optimal intervention domain in which diffusion can be effectively contained at minimal regulatory cost. In this way, our COSREF bridges structure and dynamics, offering a quantitative foundation for designing stable, efficient, and resilient regulation strategies in complex social systems.

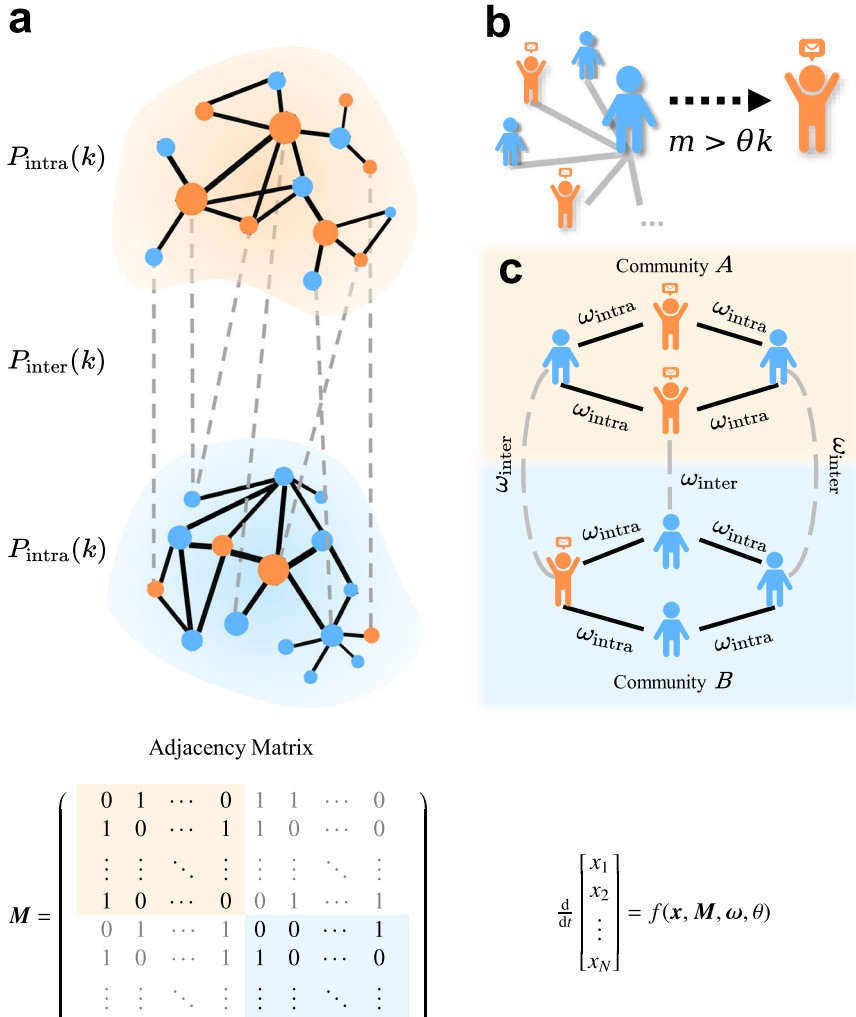

**Fig. 1 | Schematic of the community structure-regulation coupling framework (COSREF). a** A modular network comprising two communities, each characterized by intra-community and inter-community degree distributions, $P_{intra}(k)$ and $P_{inter}(k)$. The network topology is represented by the adjacency matrix $M$, where entries indicate connections between nodes. **b** Illustration of the threshold-based contagion mechanism: a susceptible (blue) individual with $k$ contacts adopts the information once the number of adopted neighbors $m$ exceeds the threshold $\theta k$. **c** Parametrization of interventions. The parameters ($\omega_{intra}$, $\omega_{inter}$) denote the effective transmissibilities that modulate transmission within and between communities, respectively, thereby acting as inverse measures of intervention stringency. The collective dynamics of information adoption are governed by a differential equation, where **x** represents the state vector of all nodes.

## Results

### Model

Our COSREF is illustrated in Fig. 1. A network is divided into two (or more) modules; for clarity, we consider two representative modules, $A$ and $B$, which can be interpreted as distinct communities such as social groups or online platforms (Fig. 1a). Links within the same module are defined as intra-links with the degree distribution $P_{intra}(k_{intra})$, and links connecting different modules as inter-links with the degree distribution $P_{inter}(k_{inter})$. The corresponding average degrees are $z_{intra}$ and $z_{inter}$, and the total average degree is $z$. We define the mixing parameter as:

$$\mu = \frac{z_{inter}}{z_{inter} + z_{intra}} = \frac{z_{inter}}{z}. \tag{1}$$

Small (large) values of $\mu$ correspond to networks with strong (weak) modular structure, whereas $\mu = 0.5$ represents a random-like connectivity pattern; explicitly, $\mu = 0$ corresponds to fully isolated modules (all intra-connections) and $\mu = 1$ corresponds to a bipartite graph with no community structure (all inter-connections). The network's community structure can also be represented by an adjacency matrix $\boldsymbol{M}$ (Fig. 1a).

Information spreading follows the susceptible-adopted dynamics, adapted from the classical susceptible-infected model without recovery, to approximate early-stage rumor propagation. Each node $i$ is in one of two states: $x_i = 0$ (susceptible) or $x_i = 1$ (adopted). Accordingly, the global adoption fraction $\rho = \frac{1}{N} \sum_i x_i$ serves as the order parameter describing the macroscopic diffusion state. Initially, a small fraction $\rho_0$ of adopters is randomly placed in one module $A$ ($B$), while all others remain susceptible. At each synchronous update, node $i$ with degree $k$ adopts according to a threshold rule[29], schematically shown in Fig. 1b,

$$\mathcal{R}(\boldsymbol{m}, \boldsymbol{\omega}, \theta, k) = \begin{cases} 1, & \boldsymbol{m} \cdot \boldsymbol{\omega} > \theta k, \\ 0, & \text{otherwise}. \end{cases} \tag{2}$$

Here, $\boldsymbol{m} = (m_{intra}, m_{inter})$ is the vector of adopted neighbors within and between modules, and $\boldsymbol{\omega} = (\omega_{intra}, \omega_{inter})$ denotes the corresponding effective transmissibility (Conversely, $(1 - \omega_{intra}, 1 - \omega_{inter})$ means the regulatory intensity). Lower $\omega$ values indicate stricter regulation, with $\omega = 0$ corresponding to complete suppression and $\omega = 1$ to unrestricted transmission. The threshold $\theta \in [0, 1]$ represents adoption sensitivity[30]: smaller $\theta$ favors easier adoption, whereas larger $\theta$ requires stronger social reinforcement.

The regulation parameters are encoded in $\boldsymbol{\omega}$ and act within and across communities as depicted in Fig. 1c: community-level moderation or algorithmic throttling primarily reduces $\omega_{intra}$, while cross-platform or inter-community filters target $\omega_{inter}$. The dot product $\boldsymbol{m} \cdot \boldsymbol{\omega}$ represents the combined regulated social influence from intra- and inter-community sources. In the absence of restrictions ($\omega_{intra} = \omega_{inter} = 1$), Eq. (2) reduces to the classical linear threshold model, widely applied in social contagion[23,29,31] and epidemic spreading[32].

The collective evolution of information adoption across all nodes can be expressed in compact form as

$$\frac{d\boldsymbol{x}}{dt} = f(\boldsymbol{x}, \boldsymbol{M}, \boldsymbol{\omega}, \theta), \tag{3}$$

where $\boldsymbol{x} = (x_1, x_2, \ldots, x_N)^{\mathsf{T}}$ denotes the state vector of node adoption, and $\boldsymbol{M}$ encodes intra- and inter-community connectivity (Fig. 1c). This general formulation highlights that global diffusion behaviour emerges from the coupling between the network topology $\boldsymbol{M}$ and regulation parameters ($\omega_{intra}, \omega_{inter}$), forming the theoretical foundation of our COSREF.

We compute the final adoption density $\rho_\infty$ using the tree-like (TL) approximation (see Methods), and validate its agreement with numerical simulations. This approach reveals how community structure and regulatory strength govern the system-level diffusion regimes.

Our framework builds on and extends the classical threshold tradition initiated by Granovetter's model of collective behaviour with heterogeneous adoption thresholds[31]. It is also consistent with the "strength of weak ties" perspective, in which cross-community bridges facilitate system-wide reach[33,34]. When the effective transmissibilities are equal, $\omega_{intra} = \omega_{inter}$, the model no longer distinguishes between within-community and between-community transmission. In that case, the dynamics reduce to a threshold process without differential regulation, governed by a single effective transmission parameter. Our framework therefore recovers the classical setting in the absence of community-specific regulation, while extending it to the case of distinct intra- and inter-community interventions.

Departing from this diagonal in the parameter space ($\omega_{intra} \neq \omega_{inter}$) decouples local reinforcement from cross-community bridging. This exposes regions where weak ties ($\omega_{inter}$) enable global cascades, and regions where strong intra-community cohesion ($\omega_{intra}$) traps diffusion. The explicit separation between community structure and regulation ($\omega_{intra}, \omega_{inter}$) clarifies when small changes in either "weak ties" or "strong ties" push the system across abrupt boundaries separating no diffusion, localized spread, and global cascades. This interpretation lays the conceptual groundwork for the regime diagrams presented in the next section.

### Regimes of diffusion and phase transitions

Varying the effective transmissibilities ($\omega_{intra}, \omega_{inter}$) and the mixing parameter $\mu$ yields three distinct diffusion regimes, as shown in Fig. 2. Unless stated otherwise, we fix the adoption threshold $\theta = 0.25$ and assume Poisson degree distributions for both intra- and inter-links,

$P_{intra}(k_{intra}) = \frac{z_{intra}^{k_{intra}} e^{-z_{intra}}}{k_{intra}!}$ and $P_{inter}(k_{inter}) = \frac{z_{inter}^{k_{inter}} e^{-z_{inter}}}{k_{inter}!}$.

i) We first consider the case $\omega_{inter} = 1$, representing unrestricted inter-community diffusion. The steady-state adoption density $\rho_\infty$ exhibits three distinct phases (Fig. 2a): a non-diffusion phase (yellow), where the spread remains confined to initially activated nodes ($\rho_\infty \approx \rho_0$); a localized diffusion phase (cyan), in which adoption saturates within the originating community ($\rho_\infty \approx 0.5$); and a global diffusion phase (dark blue), where adoption becomes system-wide ($\rho_\infty \approx 1$). The transitions between these regimes are abrupt, reflecting the nonlinear threshold response characteristic of complex contagion processes. Larger $\mu$ values (weaker community structure) facilitate cross-community cascades, whereas smaller values of $\mu$ strengthen local containment within the initially activated module.

To further elucidate the abruptness of these transitions, we fix $\omega_{intra} = 0.7$ (horizontal dashed line in Fig. 2a) and examine $\rho_\infty(\mu)$ (Fig. 2b). The TL approximation accurately reproduces the sharp phase transition observed in simulations. These analyses consistently reveal a parameter window of $\mu$ in which the information spread is maximally suppressed, a result confirmed by simulations and the analytical method.

ii) We next consider the case $\omega_{intra} = 1$, corresponding to unrestricted intra-community spreading. The phase diagram (Fig. 2d) again reveals the three canonical regimes, but now a moderate $\mu$ minimizes global diffusion. Fixing $\omega_{inter} = 0.7$ (Fig. 2e) shows that there exists an optimal range of $\mu$ at which information propagation is most effectively inhibited. This non-monotonic relationship highlights a structural sweet spot where limited interconnectivity hinders cross-community reinforcement. This strongly non-monotonic behaviour arises from the competition between local reinforcement and global connectivity. As $\mu$ increases, dense intra-community links that sustain complex contagion are progressively replaced by inter-community bridges. This structural dilution initially suppresses diffusion because

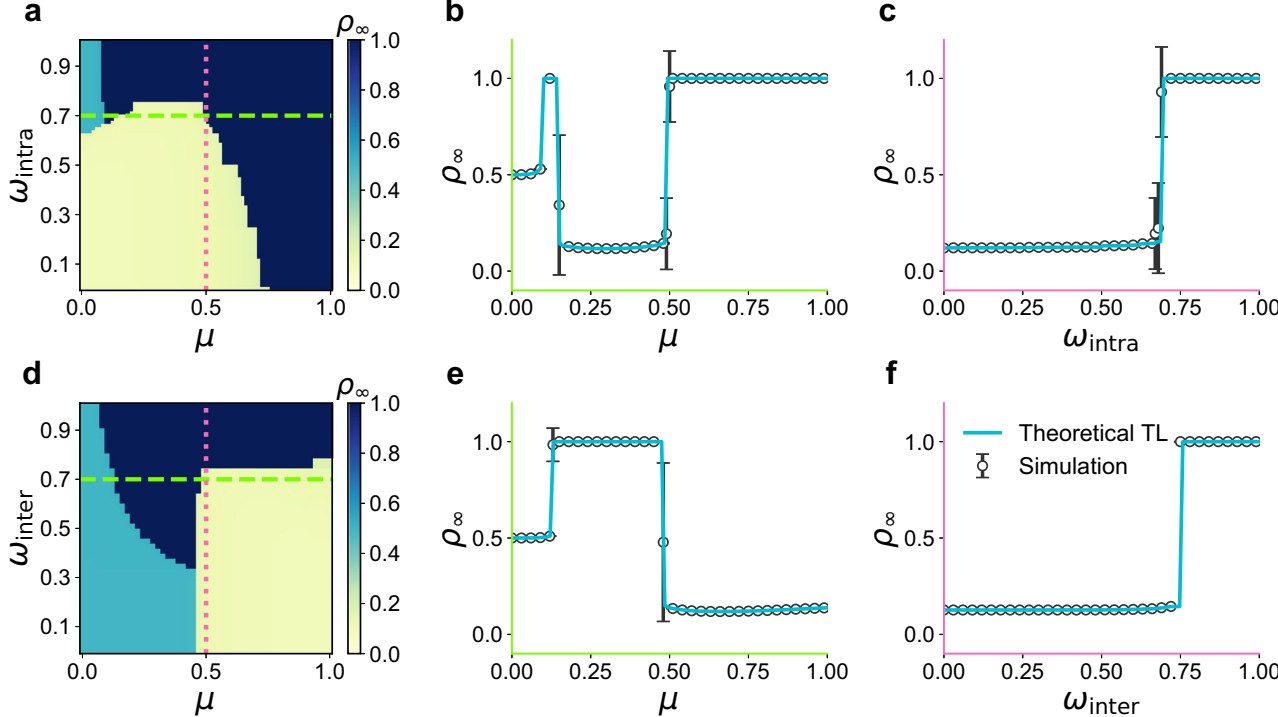

**Fig. 2 | Regimes of diffusion and phase transitions under varying network community structure. a**, Phase diagram of the final adoption density $\rho_\infty$ obtained from the tree-like (TL) approximation without cross-community regulation ($\omega_{inter} = 1$). Three regimes are identified: a non-diffusion state (yellow), a localized diffusion phase confined to community $A$ (cyan), and a global diffusion phase (dark blue). The horizontal dashed line indicates $\omega_{intra} = 0.7$ (profile shown in **b**), and the vertical dotted line marks $\mu = 0.5$ (profile shown in **c**). **b** Cross-section of panel (a) along the horizontal dashed line, showing $\rho_\infty$ as a function of $\mu$. **c**, Cross-section of panel (a) along the vertical dotted line, showing $\rho_\infty$ versus $\omega_{intra}$. **d** Same as (a) but with unrestricted intra-community transmission ($\omega_{intra} = 1$). The horizontal dashed and vertical dotted lines indicate the parameter slices presented in panels (**e**) and (**f**) respectively. **e-f**, Cross-sections corresponding to (d), illustrating the transitions between diffusion regimes. Theoretical TL (blue solid line) results match the simulations (black circles with error bars indicating standard deviations). All simulations are performed with $z = 15$, $\theta = 0.25$, $\rho_0 = 0.1$, and $N = 2 \times 10^5$.

local reinforcement weakens before cross-community connectivity becomes strong enough to sustain global spreading. A more systematic exploration of this behaviour is provided in the Supplementary Material (Supplementary Figs. 1 and 2).

### Optimal intervention domain

While strong regulation can suppress diffusion, it also entails substantial socio-economic costs. We therefore seek optimal control strategies that minimize diffusion with the least possible intervention. Our analysis proceeds in three steps: (i) quantify the impact of $\omega_{intra}$, $\omega_{inter}$, and $\mu$ on diffusion; (ii) introduce a cost function for intervention intensity; and (iii) determine the minimal-cost configuration within the controllable parameter space.

At fixed $\mu = 0.5$ and $\omega_{inter} = 1$, an abrupt transition from the non-diffusion to global diffusion phase occurs at $\omega_{intra} \approx 0.7$ (Fig. 2c). Analogous discontinuities are observed when varying $\omega_{inter}$ while fixing $\omega_{intra} = 1$ (Fig. 2f). Such sharp phase boundaries persist across broad parameter ranges (Supplementary Figs. 2a–d), confirming that the system undergoes abrupt transitions with respect to both intra- and inter-community transmissibility.

To identify the non-diffusion domain, we calculate phase diagrams via the TL approximation and numerical simulations for varying mixing parameters $\mu = 0.2, 0.4, 0.6$, and $0.8$ with $\theta = 0.1$ (Fig. 3). The yellow region, denoted as $\mathcal{U}_{free}$, represents parameter combinations where diffusion fails to percolate. Within this region, we formulate an optimization problem,

$$\boldsymbol{\omega}_o = \arg\min_{\boldsymbol{\omega} \in \mathcal{U}_{free}} \mathcal{F}(\boldsymbol{\omega}), \tag{4}$$

where $\mathcal{F}(\boldsymbol{\omega})$ quantifies the socio-economic cost of regulation,

$$\mathcal{F}(\boldsymbol{\omega}) = \frac{e^{-(\omega_{intra} + \omega_{inter})} - e^{-2}}{1 - e^{-2}}. \tag{5}$$

This monotonically decreasing function ranges from $\mathcal{F}(\boldsymbol{\omega}) = 1$ (maximal restriction, $1 - \omega_{intra} = 1 - \omega_{inter} = 1$) to $\mathcal{F}(\boldsymbol{\omega}) = 0$ (no restriction, $1 - \omega_{intra} = 1 - \omega_{inter} = 0$). The exponential form is specifically chosen to capture a realistic socio-economic scenario in which intervention costs escalate nonlinearly as regulation approaches complete suppression, thus $1 - \omega \to 1$. Moderate interventions are relatively inexpensive, whereas enforcing a near-total blockage of transmission requires disproportionately greater effort and resources. Although this form provides a mathematically convenient, bounded, and convex representation of regulatory burden, our main findings do not rely on its specific mathematical shape. Robustness tests using alternative cost definitions, such as linear and quadratic forms (see Supplementary Fig. 4), yield a consistent non-monotonic profile for the optimal cost. This confirms that the observed "sweet spot" at intermediate levels of community structure is a robust topological feature, independent of the exact cost function, provided that the cost increases strictly with regulatory strength.

Combining Eqs. (5) and (2) yields $\boldsymbol{\omega}_o$ as the optimal control configuration. Because the cost function depends symmetrically on $\omega_{intra}$ and $\omega_{inter}$, its iso-cost contours are straight lines with a slope of $-1$ (see Supplementary Fig. 16). The optimal solution therefore occurs at the tangency between these contours and the boundary of the controllable region, which geometrically determines the minimal-cost intervention strategy. This construction reveals the mechanism behind

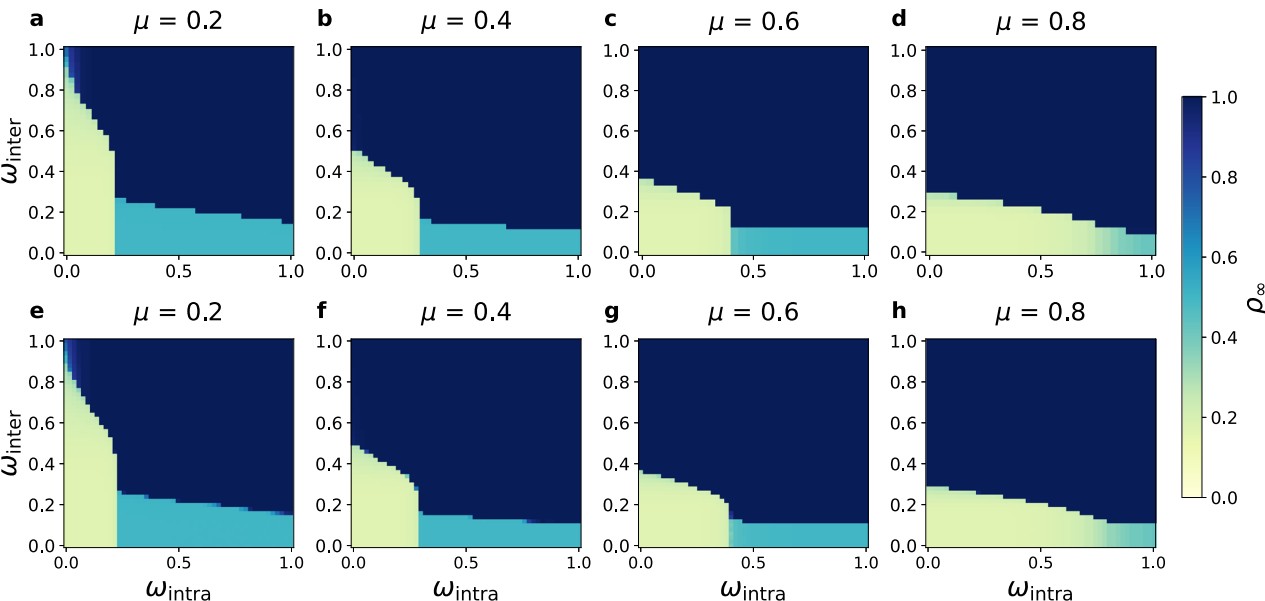

**Fig. 3 | Optimal intervention domain under varying network community structure. a–d** Phase diagrams of the final adoption density $\rho_\infty$ within parameters space $\boldsymbol{\omega} = (\omega_{intra}, \omega_{inter})$ obtained from the TL approximation, and **e–h** the corresponding simulation results for network mixing parameters $\mu = 0.2, 0.4, 0.6,$ and 0.8, respectively. Both intra- and inter-community degree distributions follow Poisson statistics, $P_{intra}(k) = \frac{z_{intra}^k e^{-z_{intra}}}{k!}$ and $P_{inter}(k) = \frac{z_{inter}^k e^{-z_{inter}}}{k!}$. Yellow regions denote the non-diffusion phase, corresponding to the controllable regime $\mathcal{U}_{free}$ where information spreading is effectively contained. The close agreement between TL approximations and simulations confirms the robustness of the theoretical framework across community structures. All simulations use $\theta = 0.1, \rho_0 = 0.17, z = 20$, and $N = 200,000$ (simulations).

the optimal control configuration: (i) For networks with pronounced community structure ($\mu < 0.5$), communities are tightly connected internally, and diffusion is primarily sustained by intra-community reinforcement. In this regime, the controllable boundary is steep, and the optimal point favors stronger intra-community regulatory strength (lower $\omega_{intra}$). (ii) For networks with weak community structure ($\mu > 0.5$), inter-community connections become more abundant and drive diffusion via cross-community bridges. The controllable boundary becomes flatter, shifting the optimal point toward stronger inter-community regulatory strength (lower $\omega_{inter}$). The resulting optimal cost $\mathcal{F}(\boldsymbol{\omega}_o)$ exhibits a symmetric unimodal distribution, peaking at $\mu \approx 0.5$, implying that networks with strong macroscopic organization (very small or very large $\mu$) are easier to control and incur lower loss, whereas networks with moderate community structure require stronger interventions, thus higher cost. This finding is robust across variations in $N, z, \theta$, and $\rho_0$ (Figs. 2, 3 and 4, Supplementary Figs. 3 and 4).

### Robustness and validation across real-world networks

To examine how community structure and intervention processes reshape the sensitivity of diffusion to regulatory parameters, we systematically evaluate the robustness across diverse network architectures (including Poisson, Scale-Free, and the analyzed real-world networks). Replacing the Poisson degree distributions with heterogeneous ones-such as fully scale-free (SF-SF-SF, thus, both intra- and inter-module connections follow power-law distributions) structure in Fig. 4 and hybrid ER-SF-ER topologies (modules are internally Erdős-Rényi networks, with inter-module connections following a power-law distribution)-produces qualitatively consistent results (Supplementary Figs. 5–7). Although scale-free networks exhibit slightly smaller final adoption densities ($\rho_\infty \approx 0.8$) in the global diffusion phase due to the prevalence of low-degree nodes, the position and extent of the controllable domain $\mathcal{U}_{free}$ remain nearly unchanged. A similar phase behaviour is observed in random-regular (RR-RR-RR) networks (Supplementary Fig. 8), confirming that the observed transitions arise from the intrinsic interdependence

between community structure and regulation rather than from artifacts of specific connectivity patterns.

Complex contagion models, including Eq. (2), typically assume that activation thresholds scale with node degree $k$. However, real diffusion processes may instead follow a constant or absolute activation rule, independent of the degree. To test this possibility, we also examined a constant-threshold contagion mechanism,

$$\mathcal{R}(\boldsymbol{m}, \boldsymbol{\omega}, C) = \begin{cases} 1, & \boldsymbol{m} \cdot \boldsymbol{\omega} > C, \\ 0, & \text{otherwise,} \end{cases} \tag{6}$$

where $C$ denotes a fixed activation constant. This variant, conceptually analogous to Boltzmann-like activation, captures processes where adoption depends on an absolute stimulus rather than proportional influence, for instance, aerosol-based spreading or collective decision cascades[35]. As shown in Supplementary Fig. 9, TL approximation results for $\mu = 0.2, 0.8$ and $C = 2, 3$ closely match those of the proportional-threshold case, demonstrating that the no-diffusion regimes are insensitive to the specific threshold definition.

Finally, we validate the COSREF on three real-world empirical social networks, i.e., Friendster, YouTube, and Orkut, sourced from the Stanford Network Analysis Project[36]. In each network, two interconnected communities with the largest number of nodes are selected (visualized as Figs. 5a-c). Unlike idealized modular networks, the empirical communities exhibit structural asymmetry: the average intra-community degree differs between them, which we denote as $z_A$ and $z_B$, respectively. The corresponding empirical phase diagrams (Figs. 5d-f) reveal the same three regimes observed in the COSREF: a non-diffusion phase ($\rho_\infty \approx \rho_0$), a localized diffusion phase ($\rho_\infty \approx 0.5$), and a global diffusion phase ($\rho_\infty \approx 1$). Smaller ($\omega_{intra}, \omega_{inter}$) representing stricter moderation, effectively suppress information propagation, especially at moderate community structure ($\mu \approx 0.5$). Despite strong heterogeneity and stochastic noise, the transition profiles in Figs. 5g–i remain abrupt and closely follow the theoretical predictions. The slight smoothing of transitions across empirical datasets can be attributed to heterogeneity in degree and community size (see Supplementary

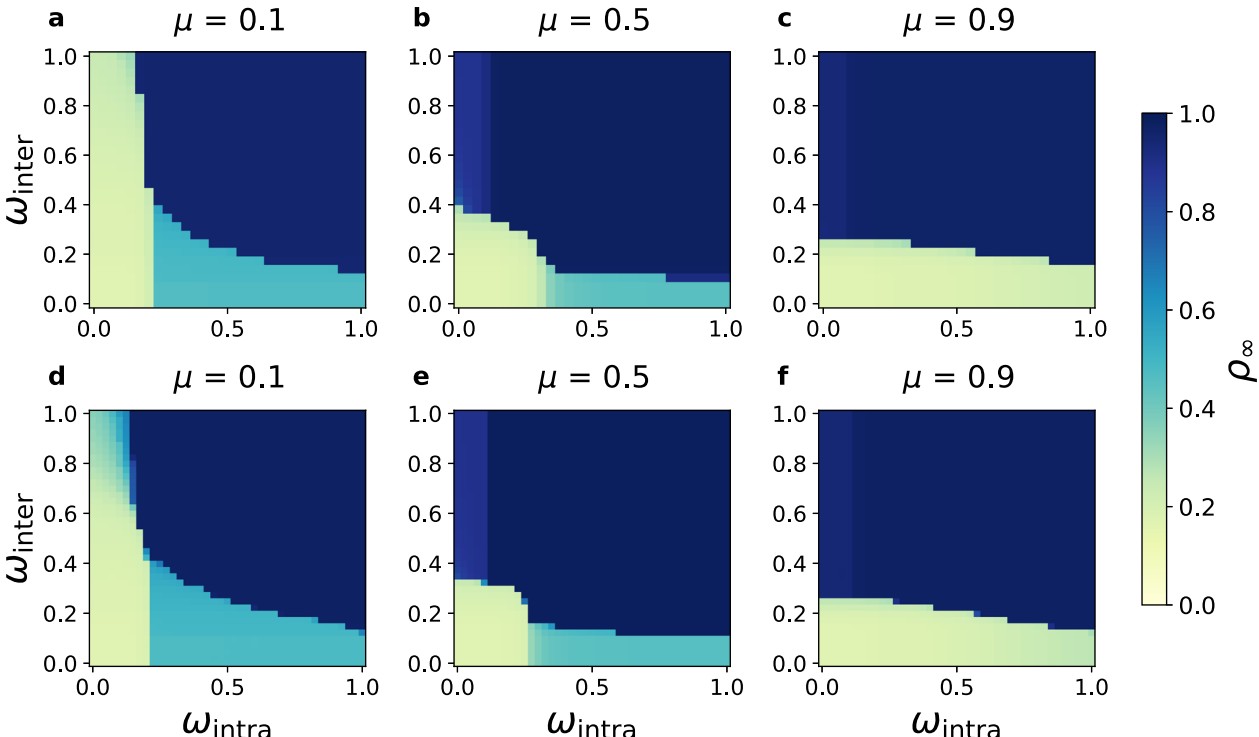

**Fig. 4 | Optimal intervention in SF-SF-SF networks.** Phase diagrams obtained via the TL approximation are shown in panels **a–c** and simulation results are shown in panels **d–f** for mixing parameters $\mu$ = 0.1, 0.5, and 0.9, respectively. Both intra- and inter-community degree distributions follow power laws, $P_{intra}(k_{intra}) \sim k_{intra}^{-\lambda_{intra}}$ and $P_{inter}(k_{inter}) \sim k_{inter}^{-\lambda_{inter}}$. Results demonstrate that the community structure-regulation coupling remains robust in fully heterogeneous (SF-SF-SF) networks, with theoretical and simulated phase boundaries in close agreement. Parameters: $\theta$ = 0.1, $\rho_0$ = 0.17, $\lambda_{intra}$ = 3, $\lambda_{inter}$ = 3, $z$ = 20, and $N$ = 200, 000 (simulations).

Fig. 10). Collectively, these results support the robustness across investigated topologies of the COSREF.

A key limitation of empirical networks is that the mixing parameter $\mu$ is fixed by the observed topology, preventing direct exploration of its influence on diffusion regimes. To overcome this, we apply a degree-preserving edge reshuffling procedure to each of the three real-world networks (Friendster, YouTube, and Orkut), systematically increasing or decreasing $\mu$ while retaining the original degree distribution. The resulting phase diagrams and cross-sections (Supplementary Figs. 12–14) confirm that the three canonical diffusion regimes-non-diffusion, localized, and global-persist across the full range of reshuffled $\mu$ values, and that the intervention domain observed in synthetic models is faithfully reproduced on real topologies. These reshuffling experiments thus provide direct empirical evidence that the community structure-regulation coupling is not an artifact of idealized network models but emerges robustly from the interplay between community structure and regulatory parameters in real social networks.

## Discussion

Information dissemination in social systems is fundamentally shaped by reinforcement effects: individuals are more likely to adopt an idea after repeated exposure within their social circles. The proposed COSREF captures this collective reinforcement through a threshold-based contagion model. It provides a minimal yet general representation of complex information diffusion. Unlike models assuming homogeneous mixing or single-layer interactions, it explicitly integrates modular community structure and regulatory parameters, revealing how the balance between intra- and inter-community transmission governs global spreading outcomes.

Our COSREF results demonstrate that community structure can both suppress and amplify diffusion, depending on the intervention regime. This duality produces discontinuous transitions between local and global adoption states, a hallmark of cooperative contagion dynamics. Simulations on real-world social networks further confirm that the observed transitions and control regimes are robust across investigated topologies. Crucially, the identified controllable domain offers a theoretical basis for optimizing information moderation, enabling effective containment with minimal intervention cost.

This framework also extends naturally to recurrent contagion dynamics. By adopting a Susceptible-Infected-Susceptible formulation that allows reinfection[37], we derived degree-based equations (see Supplementary Material) and found qualitatively similar intervention domains and abrupt phase transitions. This suggests that the coupling between community structure and effective transmissibility represents a general organizing principle underlying both transient and cyclic diffusion processes in social and informational systems.

Our findings indicate that modular networks offer structural opportunities for implementing efficient, low-cost control strategies against misinformation or rumor propagation. Moreover, the COSREF is not limited to two-module systems but can be generalized to multi-modular configurations. In such cases, the threshold rules in Eqs. (2) and (6) can be reformulated in matrix form, where the adopted neighbor counts $\boldsymbol{m}$ and transmissibility coefficients $\boldsymbol{\omega}$ are expressed as block matrices. As an explicit demonstration, we instantiate the general TL framework for a three-module system and validate the resulting predictions against Monte Carlo simulations (see Section III in Supplementary Material, and Supplementary Fig. 15); the excellent theory-simulation agreement confirms that the community structure-regulation coupling identified in the two-module case extends naturally to multi-module topologies. The diagonal blocks represent intra-module influence, while off-diagonal terms capture heterogeneous inter-module interactions. This generalization enables the modeling of richer cross-regional and cross-platform contagion phenomena in complex social ecosystems.

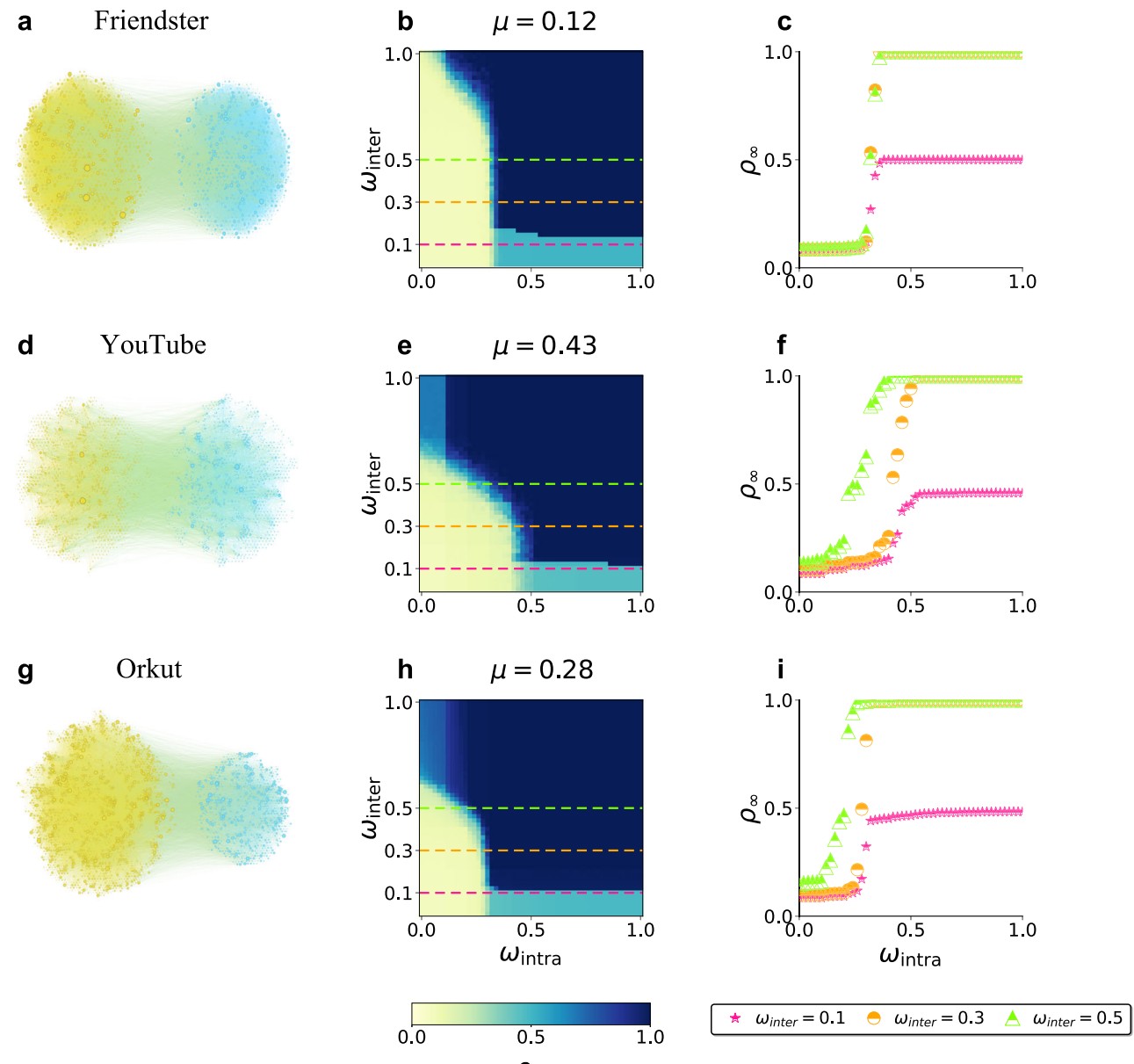

**Fig. 5 | Empirical validation of optimal information diffusion control on real-world social networks. a–c** Representative network structures of the Friendster, YouTube, and Orkut social networks, respectively. **d–f** Corresponding phase diagrams of the final adoption density $\rho_\infty$ as functions of the intra- and inter-community effective transmissibilities, $\omega_{intra}$ and $\omega_{inter}$, exhibiting the same three characteristic regimes predicted by theory: non-diffusion (yellow), localized diffusion (cyan), and global diffusion (dark blue). **g–i** Cross-sections of $\rho_\infty$ versus $\omega_{intra}$ at fixed $\omega_{inter}$ values (0.1, 0.3, and 0.5), showing sharp, abrupt transitions consistent with theoretical predictions. All networks are obtained from the Stanford Network Analysis Project (SNAP): Friendster ($N = 2368$, $\mu = 0.12$, $z_A = 15$, $z_B = 10$), YouTube ($N = 3269$, $\mu = 0.43$, $z_A = 1$, $z_B = 1$), and Orkut ($N = 4621$, $\mu = 0.28$, $z_A = 5$, $z_B = 1$). Parameters: $\theta = 0.1$, $\rho_0 = 0.17$.

Beyond information diffusion, this interdependence between community structure and the effectiveness of regulatory parameters may also be relevant to a broader class of spreading and cascading phenomena. Analogous dynamics arise in risk and disaster propagation across coupled natural and social systems-such as the cascading failure of interdependent infrastructures[38,39], the diffusion of financial crises, or the spatial spread of climate-related hazards[40]. Community structure and regulation coupling jointly determine system resilience and recovery efficiency, paralleling the controllability domains identified here. Thus, the proposed COSREF provides a general theoretical foundation for understanding and managing multi-sectoral risks in complex socio-environmental networks.

While our framework introduces $\omega$ as a generalized mathematical representation of reduced transmissibility, this parameter can be related to practical interventions in several contexts. In online information systems, decreasing $\omega_{inter}$ may correspond to reducing cross-community exposure, for example through downranking, friction, or limits on recommendation pathways that connect otherwise weakly linked groups. Decreasing $\omega_{intra}$ may instead represent moderation or exposure reduction within closely connected communities. In epidemic or mobility-driven spreading processes, the same distinction may correspond to local contact reduction within communities versus restrictions on movement between geographic or demographic groups. These examples should not be interpreted as direct prescriptions, since real implementation would require temporal data, behavioural heterogeneity, platform-specific constraints, and ethical considerations. Rather, the framework provides a theoretical way to compare how intervention effort might be allocated between within-

community and between-community transmission channels under different community structures.

By linking community structure, regulation, and contagion dynamics within a single coherent framework, this study establishes a foundation for designing adaptive moderation and mitigation strategies, advancing the understanding of stability, efficiency, and resilience across diverse real-world systems.

Several aspects not addressed in the present study point to promising directions for future research. First, our framework considers how network structure shapes spreading dynamics on a static topology; however, in real social systems, the spreading process itself can reshape the underlying network, for example, through opinion-dependent link rewiring or adaptive contact patterns. Incorporating such co-evolutionary dynamics between structure and process would provide a more realistic description of information ecosystems. Second, the threshold model employed here is a stylized representation of social reinforcement; real adoption processes involve richer decision-making mechanisms influenced by individual heterogeneity, information quality, and contextual factors. Extending the framework to accommodate more complex contagion rules is a natural next step. Third, the abrupt transitions observed in our model may reflect either genuinely discontinuous behaviour or continuous transitions with explosive characteristics (sharp transitions and atypical finite-size scaling). A rigorous characterization of the transition order through finite-size scaling analysis and estimation of critical exponents would deepen the understanding of the underlying critical phenomena and constitutes an important direction for future work.

## Methods
### Data
We employ three real-world social network datasets from the Stanford Network Analysis Project (SNAP)[36], all drawn from the "Networks with Ground-Truth Communities" collection. Each dataset provides verified community labels that are essential for quantitative validation. The networks are: (i) Friendster - a large-scale friendship network containing 65,608,366 nodes and 1,806,067,135 edges; (ii) YouTube - a video-sharing platform network with 1,134,890 nodes and 2,987,624 edges; and (iii) Orkut - a social networking service comprising 3,072,441 nodes and 117,185,083 edges. For YouTube and Orkut, ground-truth communities correspond to user-defined groups within each platform.

Since sparse inter-community connections can obscure phase-transition structures and weaken validation, we carefully selected pairs of communities exhibiting substantial inter-community connectivity. This ensures clear phase-transition patterns that are critical for testing theoretical predictions. We focus on community pairs with non-negligible inter-community connectivity. When inter-community links are extremely sparse, the two communities behave as nearly isolated subsystems, and the effect of inter-community transmission becomes negligible. In such cases, the spreading dynamics are dominated by intra-community processes, making it difficult to meaningfully assess the role of inter-community regulation. Our analysis therefore focuses on coupled systems in which both intra- and inter-community transmission channels contribute to the dynamics. The selected community pairs contain nodes belonging to both groups. Although our theoretical model does not explicitly account for overlapping nodes, assigning shared nodes to either community is theoretically valid and allows direct application of the model. We adopt two preprocessing strategies: (i) a biased assignment, in which overlapping nodes are preferentially assigned to the smaller community to balance sizes; and (ii) a random assignment, where overlapping nodes are distributed randomly. Here we use the biased assignment method and select two

representative communities from each dataset, containing 2368, 3269, and 4621 nodes, corresponding to $\mu$ = 0.12, 0.43, and 0.28, respectively. Experimental validation (Supplementary Fig. 11) confirms that both assignment schemes yield nearly identical optimal-intervention phase diagrams, demonstrating that preprocessing choices do not qualitatively affect the dynamical behaviour or intervention domains.

### Theoretical formalism
Here, we propose a general form for the Tree-Like (TL) approximation of the $n$-community solution. This framework assumes that the network structure is locally tree-like, such that nodes at hierarchy level $l$ are influenced solely by those at level $l − 1$. Focusing on the $l$-th community, the fraction of adopted nodes is computed through the auxiliary variable $y_l^I$. The connectivity of any chosen site in community $I$ is described by the degree vector $\mathbf{k} = (k_1, k_2, \cdots, k_n)$, where the element $k_i$ denotes the degree of the site connecting to community $i$. A similar definition applies to the vector of adopted neighbors, $\mathbf{m}$.

Accordingly, the evolution of $y_l^I$ is given by:

$$
\begin{aligned}
y_{l+1}^I = \rho_0^I + \frac{(1 - \rho_0^I)}{z} \\
\times \sum_{\{\mathbf{k}\}} \Bigg\{ P(\mathbf{k}) \times \sum_{i=1}^{n} \Bigg[ k_i \sum_{m_i=0}^{k_i-1} \mathcal{B}(m_i, k_i - 1, y_l^i) \sum_{\{\mathbf{m}\}\setminus\{m_i\}}^{\{\mathbf{k}\}\setminus\{k_i\}} \prod_{j \neq i} \mathcal{B}(m_j, k_j, y_l^j) \\
\times \mathcal{R}(\mathbf{m}, \boldsymbol{\omega}, \theta, \mathbf{k} - 1 \times \widehat{\mathbf{e}}_I) \Bigg] \Bigg\},
\end{aligned}
$$
(7)

leading to the steady-state fraction of adopted nodes:

$$
\rho_\infty^I = \rho_0^I + (1 - \rho_0^I) \sum_{\mathbf{k}}^{\infty} P(\mathbf{k}) \sum_{\{\mathbf{m}\}}^{\{\mathbf{k}\}} \prod_j \mathcal{B}(m_j, k_j, y_\infty^j) \times \mathcal{R}(\mathbf{m}, \boldsymbol{\omega}, \theta, \mathbf{k}).
$$
(8)

Here, $\rho_0^I$ denotes the initial adoption ratio in the $I$-th community, and $P(\mathbf{k}) = \prod_i P_i(k_i)$ represents the joint probability density of degrees across all communities. The summation $\sum_{\{\mathbf{k}\}} = \sum_{k_1} \cdots \sum_{k_n}$ encompasses all possible sets of parameters $\{k_1, \cdots, k_n\}$, and the notation $\{\mathbf{m}\} \setminus \{m_i\}$ refers to the set of all parameters in $\mathbf{m}$ excluding $m_i$. The binomial term $\mathcal{B}(m_j, k_j, y_l^j) = \binom{k_j}{m_j} (y_l^j)^{m_j} (1 - y_l^j)^{k_j - m_j}$ describes the probability that $m_j$ of the $k_j$ links in community $j$ connect to adopted nodes. To account for the cavity effect, the threshold rule $\mathcal{R}(\mathbf{m}, \boldsymbol{\omega}, \theta, \mathbf{k} - 1 \times \widehat{\mathbf{e}}_I)$ is adjusted by decrementing the degree in the $I$-th community by 1 (represented by $-1 \times \widehat{\mathbf{e}}_I$).

For the specific case of $n = 2$, comprising communities $A$ and $B$, we take community $A$ as a representative example. The degree vector is defined as $\mathbf{k} = (k_{\text{intra}}, k_{\text{inter}})$, describing the number of intra-community links $k_{\text{intra}}$ and inter-community links $k_{\text{inter}}$, respectively. An analogous definition applies to $\mathbf{m}$. In this scenario, the evolution equation simplifies to:

$$
\begin{aligned}
y_{l+1}^A = \rho_0^A + (1 - \rho_0^A) \sum_{k_{\text{intra}}, k_{\text{inter}}}^{\infty} \frac{k_{\text{intra}} + k_{\text{inter}}}{z} P_{\text{intra}}(k_{\text{intra}}) P_{\text{inter}}(k_{\text{inter}}) \\
\times \Bigg\{ \frac{k_{\text{intra}}}{k_{\text{intra}} + k_{\text{inter}}} \sum_{m_{\text{intra}}=0}^{k_{\text{intra}}-1} \sum_{m_{\text{inter}}=0}^{k_{\text{inter}}} \mathcal{B}(m_{\text{intra}}, k_{\text{intra}} - 1, y_l^A) \mathcal{B}(m_{\text{inter}}, k_{\text{inter}}, y_l^B) \\
+ \frac{k_{\text{inter}}}{k_{\text{intra}} + k_{\text{inter}}} \sum_{m_{\text{intra}}=0}^{k_{\text{intra}}} \sum_{m_{\text{inter}}=0}^{k_{\text{inter}}-1} \mathcal{B}(m_{\text{intra}}, k_{\text{intra}}, y_l^A) \mathcal{B}(m_{\text{inter}}, k_{\text{inter}} - 1, y_l^B) \Bigg\} \\
\times \mathcal{R}(\mathbf{m}, \boldsymbol{\omega}, \theta, k_{\text{intra}} + k_{\text{inter}} - 1),
\end{aligned}
$$
(9)

which yields the steady-state fraction of adopted nodes:

$$\rho_\infty^A = \rho_0^A + (1 - \rho_0^A) \sum_{k_{intra}, k_{inter}}^{\infty} P_{intra}(k_{intra}) P_{inter}(k_{inter})$$

$$\times \sum_{m_{intra}=0}^{k_{intra}} \sum_{m_{inter}=0}^{k_{inter}} \mathcal{B}(m_{intra}, k_{intra}, y_\infty^A) \mathcal{B}(m_{inter}, k_{inter}, y_\infty^B) \quad (10)$$

$$\times \mathcal{R}(\boldsymbol{m}, \boldsymbol{\omega}, \theta, k_{intra} + k_{inter}).$$

Analogous expressions apply to community B.

### Computational methods

All Monte Carlo numerical simulations employ the efficient Mersenne Twister MT19937 algorithm for high-quality pseudorandom number generation[41]. To account for stochastic variability inherent in Monte Carlo processes, we perform 100 independent realizations for each parameter set ($\omega_{intra}$, $\omega_{inter}$). In each realization, a modular network comprising two communities is constructed, $\rho_0 N$ initial adopters are randomly selected from community A, and information diffusion evolves according to the threshold-based dynamics governed by regulatory strengths. Ensemble averaging across 100 runs ensures statistical robustness and smooth convergence of the macroscopic observables, allowing precise identification of diffusion phase boundaries and optimal intervention domains within the COSREF.

For Scale-Free (SF) networks, we employ the configuration model[42] to generate networks with power-law degree distributions $P(k) \sim k^{-\gamma}$. The procedure is as follows: (i) generate a target degree sequence by sampling each node's degree from the power-law distribution with exponent $\gamma$; (ii) create a "stub list" where each node index appears a number of times equal to its assigned degree; (iii) randomly pair stubs to form edges, avoiding self-loops and multiple edges where possible. This method produces a random network that preserves the desired SF degree distribution while maintaining the modular structure specified by the intra- and inter-community connectivity parameters.

## Data availability

All real-world social network datasets used in this study are publicly available from the Stanford Network Analysis Project (https://snap.stanford.edu/). All numerical simulation results and theoretical calculations underlying the figures in this study can be fully recalculated using the code[43] we provide.

## Code availability

The C++ and Python codes used for the analysis are available on GitHub[43].

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

## Acknowledgements

The authors would like to thank Prof. Haijun Zhou and Prof. Yueheng Lan for useful discussions.

## Author contributions

J.N and J.F designed the research. X.J.C and J.F performed the analysis and prepared the manuscript, X.J.C, M.X, J.M, S.F, X.S.C, J.K, J.N and J.F generated research ideas and discussed results, and contributed to writing the manuscript.

## Fundings

We acknowledge the support by the National Natural Science Foundation of China (Grant No. T2525011, 42575057, 42450183, 12275020, 12135003, 12205025, 42461144209), the National Key R&D Program of China (2025YFF0517203, 2025YFF0517304). J.F. acknowledges support from the MEL Visiting Fellowship and the Fundamental Research Funds for the Central Universities. Open Access funding enabled and organized by Projekt DEAL.

## Competing interests

The authors declare no competing interests.

## Additional information

**Peer review information** : *Nature Communications* thanks the Sean Cornelius, and the other, anonymous, reviewer(s) for their contribution to the peer review of this work. A peer review file is available.

