## [Transparent Peer Review file · Nature Communications]

Community structure–regulation coupling reveals optimal information diffusion

Corresponding Author: Professor Jan Nagler

Version 0:

Reviewer comments:

Reviewer #1

(Remarks to the Author)

This manuscript considers the problem of controlling (mis)information spread in social networks. To this end, the authors introduce a so-called "Modularity-Controllability Coupling Framework" (MOCOF). Specifically, they consider a threshold-based model of contagion similar to the Watts model for rumour spread, played out on modular networks. The authors adopt control parameters that, for example, modulate the efficiency of spread within vs. between communities.

The authors find distinct phases in this inter/intra space, corresponding to no adoption (0%), localized adoption, and full adoption/contagion (100%). Per the authors, this suggests, that small adjustments to control parameters may result abrupt suppression of misinformation spread. The authors use a cost function to identify an "optimal controllability domain", in which diffusion is minimized with the least social pressure. The authors validate their findings on real-world social network. Overall, I think this is a well-done study that could be a good fit for this journal. In particular, I commend the authors for addressing a nontrivial problem in *nonlinear* control—making this work actually useful—rather than assuming linear dynamics for the sake of analytical tractability. The manuscript is also well-written on the whole.

However, I have some questions/comments about the specific choices the authors made, and the generalizability of their results. I'd like to see how the authors address these before recommending final publication:

1) Terminology

At several points (incl. figure captions) the authors refer to the "omega" parameters as "regulatory strengths". This is somewhat misleading, no? Wouldn't it be better to call these parameters "transmissibility" or "susceptibility" or similar?

It seems like it would actually be $(1 - \omega)$ or $1/\omega$ that would properly be called "regulatory strength".

2) The cost function

The authors assume a cost-of-control function that is exponential in the control parameters (the omegas). I get that this has the nice property of taking values in $[0, 1]$, but I think the justification for this particular shape of cost function is on the thin side. (Yes, it resembles a Boltzmann factor, but I'm not sure social dynamics should a priori obey the canonical ensemble or similar!)

Would the authors' main findings (e.g. the "sweet spot" of intermediate modularity) persist under other cost functions (e.g. linear, quadratic)? Does the exact cost function even matter, or is it sufficient e.g. that it be monotone increasing in both omegas?

3) The assumption of two modules

I understand that the authors focus on networks of two modules for simplicity, as there is already a lot going on with their modeling (both topology and dynamics). But of course, real social networks that are modular likely have MORE than two communities.

I think a small simulation in a system with more modules (e.g. a stochastic block model) would provide assurance that the authors' findings are universal. Even if there are some simplifying assumptions (e.g., omega-inter is the same between ALL modules, similar for omega-intra).

Such a simulation might also reveal even more rich behavior than the authors' have found with just two modules.

(Remarks on code availability)

I have only verified that the code for all simulations advertised seems to be there. I have not run any of the ipython notebook files myself.

I would advise the authors to do the following:

1) It's a bit unorthodox to store a .zip archive in a github repository. It would be better to structure the code in a directory, and keep only source (text) files, deleting any other files such as data that can be generated by the user.

2) Similarly, this repository seems quite large (~100 MB uncompressed). Much of that seems to be the data (as in 1 above), but it's also likely that your notebook files were committed with metadata/outputs (e.g. figures) included). There are also some PDF files (your figures) that, presumably are generated from the code and hence don't need to be stored in the repository.

Reviewer #2

(Remarks to the Author)

The paper tackles an important problem: controlling the spread of complex contagion with minimum cost. The work does not provide a concrete, applicable intervention strategy, rather it studies a mathematically tractable model of information diffusion and explores mechanisms that influence the dynamics. While I find the work interesting, I believe that the paper under-delivers what it promises in the introduction and that it could be strengthened by more carefully explaining the mechanisms behind the observed phenomena.

Comments:

1) The submitted article is closely related to and builds on the setup and results of Nematzadeh et al., 2014 (Ref [23]), which studies identical spreading dynamics and community structure, uses the same analytical solutions, and is the original paper pointing out the potentially counter-intuitive role of modularity. The current paper adds a control mechanism to this setup, but essentially studies the same phenomenon. I think this must be acknowledged more clearly than it is now and this Ref [23] should be cited more prominently.

2) The introduction identifies two challenges in controlling complex contagion: heterogeneity of the network structure and co-evolving dynamics and network structure. Heterogeneity is somewhat considered: although the main analysis focuses on Poisson degree distribution, some real networks are considered. However, dynamics co-evolving with network structure are not considered at all, leaving the reader wondering why it is so prominently discussed in the introduction?

3) In the abstract the authors claim that their work "provides general principles for achieving stability, efficiency, and resilience in modular systems". This should be more explicitly discussed in the paper, because right now I do not see how stability and resilience is related to the results.

4) The dynamics has three parameters: theta governing the threshold and two control knobs, the weight of inter- and intra-connections. Couldn't the number of the parameters be reduced? The parameters appear in the dynamics as $m \cdot w > \theta^k$, hence one could, for example, normalize the control weights and theta to set one of the average weights to 1.

I am not sure, but couldn't the control weights be absorbed into the degree distribution, i.e., with the substitution $z \rightarrow w \cdot z$? This would reduce the setup to the same setup studied in Ref [23].

5) The paper focuses on control, therefore, the choice of w is a central issue. This is discussed in section "Optimal controllability domain". The way I understand, there is a region of w_{inter} and w_{intra} (yellow area) where control is possible and the outcome is the same. So one would like to choose an extremal point of this region that minimizes some cost.

A possible cost function is introduced, but its consequences are not really explored. None of the figures show the optimal control strategy. For example, one could include a marker on the phase diagrams of Fig 3 showing the optimal point within the yellow region. What is the mechanism behind the control? Are there scenarios where intra control is favored over inter? Or the other way around?

6) The choice of the cost function seems a bit arbitrary. In its current form it only depends on the values of w , but not on how many links need to be controlled. Wouldn't it make more sense to study a cost that is proportional to the intervention size? I.e., $(1-w_{\text{inter}})z_{\text{inter}} + (1-w_{\text{intra}})z_{\text{intra}}$. Or something similar.

7) It is a good robustness check to consider thresholds that do not depend on the degree. It is perhaps not surprising that the authors find unchanged results for Poisson distributed networks, since in a Poisson distribution most nodes have similar degrees close to the average. I would expect to see a bigger difference for SF networks.

8) The paper would benefit from a paragraph explaining the network generation algorithm, including how the SF networks

are generated.

(Remarks on code availability)

I recommend uploading the code to github directly, not as a zip file.

Reviewer #3

(Remarks to the Author)

The authors present an interesting study aimed at characterising the interplay between complex contagion processes (i.e., threshold processes), clustering, and intra/inter diffusion parameters. The study is articulated with numerical simulations (in both synthetic and real networks), several mathematical derivations, and an overall clear/fluid narrative.

The results are interesting, but the framing and their relevance in the real world are overstated with several unsubstantiated claims. In this current form I cannot recommend this work for Nat Comm. I think that Physics journals (maybe even Nat Phys after several clarification/changes) would be better suited for these types of theoretical works.

A good place to start is this paragraph in the discussion, where the authors wrote

“Our MOCOF results demonstrate that modular organization can both suppress and amplify diffusion, depending on the controllability regime. This duality produces discontinuous transitions between local and global adoption states, a hallmark of cooperative contagion dynamics. Simulations on real-world social networks further confirm that the observed transitions and control regimes remain robust in empirical topologies. Crucially, the identified controllable domain offers a theoretical basis for optimizing information moderation, enabling effective containment with minimal intervention cost. The cost function, inspired by the Boltzmann distribution, captures how regulatory effort scales nonlinearly with intervention strength, linking statistical physics principles with social-system regulation.”

The results summarised in the first sentence are indeed interesting from a theoretical standpoint. However, the use of the word “controllability” (here as well across the paper) is a bit misleading. Indeed, what the authors do is to simulate a complex contagion process in topologies with different clustering and different intra/inter transmissibility. I am not sure whether within these settings we can speak about “controllability”. That word is typically used in the study of dynamical processes where systems dynamics are affected by active (and somewhat realistic) interventions.

The second sentence raises another important issue I have with the work presented. Indeed, the authors speak many times about first-order phase transitions. However, they do not really properly investigate the nature of such transitions and base their judgement on the figures. Are we really sure about the nature of those transitions? Do we really need to classify them in this way without an adequate investigation? As far I can tell these transitions are indeed abrupt, but as the “querelle” 15 years ago about the nature of phase transitions in explosive percolation process shows, deeper investigations are needed to really characterise these types of transitions. On a similar point, also all the claims about “universality” seems to me overselling some observations in a few topologies. Do we really need to claim universality? Do we have any theoretical support for this claim?

Regarding the third sentence, the dynamics studied are based on a simple model which is just a rough approximation of real interactions. Hence, can we really claim to have identified a method to effectively control information spreading? Also, how a method based on this paper exactly would work in practice? This appears to me yet another speculation not really supported by evidence, nor needed.

Regarding the final sentence, considering the nature of dynamics and systems under study, why do we care about minimizing a Boltzmann-like distribution?

The authors decided to use the word modularity as synonymous of clustering. I understand the choice but considering how “charged” the discussion around community detection methods based on modularity metrics is, I would suggest using the word clustering or community structure.

Figure 2 is one of the most important figures in the paper. The panels b, c, e, f are poor graphic representations, the simulation points are not visible, the lines mix and it is not clear what they show. They need to be better represented, ticker lines, better viz of the simulation points, larger plots might help. Simulations could be shown as shaded areas with median and CI. Much more work is needed here as this is a key figure of the paper. Since the MF does not really work, can we just move it to the SI to help the narration and the figure?

Interestingly, in figure 2b we see that for small values of μ (high clustering) $\rho \sim 1$, then increasing μ leads to a suppression and then increasing even more leads to another global spreading. So, we are in front of highly non-monotonic dynamics. This is a very interesting result that the authors comment, but I feel a bit more discussion about what might be responsible for this would be extremely beneficial.

In figure 2e we see another very interesting result. As $\mu \rightarrow 1$ (hence moving towards unclustered networks) we get a suppression of the spreading despite $w_{\text{intra}}=1$. What is behind this? Also in this case, more work is needed to describe

these very interesting results.

The section "Optimal controllability domain" is a bit confusing. As mentioned, before it is not clear what "control" means and how is that applied to the system. The authors say that the function $f(w)$ is used to quantify the "socio-economic" costs, but then we are shown something akin to the Boltzmann distribution. It is unclear however why this is a good choice nor where the "socio-economic" costs are considered. Outside the community of stats phys this choice, without proper motivation, will appear rather strange.

The authors speak about "modularity-controllability coupling mechanisms". However, it is still not particularly clear which mechanism they are referring to as dynamics are not linear and clearly dependent on the phase space. So, what they mean by this?

In the real networks we have only one value of μ for each graph. The authors could try to reshuffle the networks to increase/decrease μ to study the impact of this variable starting from a real topologies. It would be also good to see even just in the SI a similar figure to figure 4 for the network's models.

Why are the main results obtained using Poissonian degree distributions? They are not really a good description of real (aggregated) interaction patterns.

In the methods the authors wrote:

"Since sparse inter-community connections can obscure phase-transition structures and weaken validation, we carefully selected pairs of communities exhibiting substantial inter-community connectivity."

This appears very close to handpicking to me, which is very peculiar considering the claims of "universality" the authors make also in these networks. Could we see what happens in the randomly selected pairs of other communities?

I found particularly strange the choice to not discuss any limitations. For example, the authors consider static topologies, while we know real dynamics are subject to complex temporal patterns. None of this is considered here. The dynamics are just a simple model which approximate real interactions etc.

In figure Figure S1 (panel C) the MF solutions show a very strange peak (which appears very off the trend). What's that?

(Remarks on code availability)

Version 1:

Reviewer comments:

Reviewer #1

(Remarks to the Author)

The authors have satisfactorily addressed my comments in the first round of review. In particular, I commend them for going above-and-beyond in one, case, extending their theoretical framework to more than two modules (I had asked only about a simulation).

I recommend publication.

(Remarks on code availability)

The authors followed my advice in restructuring their GitHub repository into a more orthodox format. By my standards, the code is sufficiently "available" that a reader could use it. However, I have not taken the time to run the various notebooks, etc., to ensure they produce the intended output. I evaluate their work instead based on the figures in the manuscript and response to reviewers.

Reviewer #2

(Remarks to the Author)

The authors addressed the points that the other referees and I raised, and these changes clarified the context of the paper and made it more readable. I believe that with the added robustness checks the authors obtain close to the maximum of what can be learned from the setup. Having that said, overall much of the changes went in the direction of explaining the limitations of the work and adjusting the claims of the introduction to the actual results.

I appreciate that Nematzadeh et al. (2014) is discussed now more prominently and it is explained that the introduction of the w parameter is the main difference. I maintain that it is arguable how substantial this addition is. The manuscript investigates the same phenomena as Nematzadeh et al. (2014) with an additional parameter (that could be mathematically incorporated into the network parameters) and branding w as a control knob. The introduction still reviews previously proposed more realistic regulation strategies and points out issues of practical implementation to motivate the current work. However, the

proposed control mechanism is purely theoretical without any discussion of practical implementation.

I do believe that there is much value to theoretical models. In my opinion, however, the results of the manuscript are incremental and lack the level of significant new understanding that the authors claim.

(Remarks on code availability)

Reviewer #3

(Remarks to the Author)

The authors did a very good job in the revision. All the points raised in the first round have been addressed. I recommend the article for publication.

(Remarks on code availability)

As a general note, to improve terminological precision, we have revised the manuscript throughout to use "community structure" in place of "modularity" where appropriate, and "regulation", "regulatory", or "intervention" in place of "controllability", except where the latter is retained in a narrowly defined technical sense. Accordingly, the title has been updated to "*Community structure–regulation coupling reveals optimal information diffusion*" to reflect these refinements.

Response to Reviewer #1

General Comment:

This manuscript considers the problem of controlling (mis)information spread in social networks. To this end, the authors introduce a so-called "Modularity-Controllability Coupling Framework" (MOCOF). Specifically, they consider a threshold-based model of contagion similar to the Watts model for rumour spread, played out on modular networks. The authors adopt control parameters that, for example, modulate the efficiency of spread within vs. between communities.

The authors find distinct phases in this inter/intra space, corresponding to no adoption (0%), localized adoption, and full adoption/contagion (100%). Per the authors, this suggests, that small adjustments to control parameters may result abrupt suppression of misinformation spread. The authors use a cost function to identify an "optimal controllability domain", in which diffusion is minimized with the least social pressure. The authors validate their findings on real-world social network.

*Overall, I think this is a well-done study that could be a good fit for this journal. In particular, I commend the authors for addressing a nontrivial problem in *nonlinear* control—making this work actually useful—rather than assuming linear dynamics for the sake of analytical tractability. The manuscript is also well-written on the whole.*

Response:

Thank you for your thoughtful review and positive feedback. We appreciate the recognition of the relevance of the problem, the nonlinear control perspective adopted in our framework, and the overall clarity of the presentation.

Comment 1.0:

However, I have some questions/comments about the specific choices the authors made, and the generalizability of their results. I'd like to see how the authors address these before recommending final publication:

Response to 1.0:

Thank you very much. We have carefully addressed your specific concerns regarding terminology, model assumptions, and generalizability. Specifically, we have: (1) **Ensured consistent application of terminology to distinguish "transmissibility" (ω) from "regulatory intensity" ($1 - \omega$)**; (2) **Expanded the cost function analysis** to demonstrate that our results are robust under alternative formulations (including linear and quadratic forms), clarifying that the identified "optimal domain" reflects the structure of the diffusion boundary rather than an artifacts of the specific cost function; and (3) **Generalized the model** to support an **arbitrary number of modules** and verified that

the key qualitative behaviors persist in multi-community systems. Detailed responses to each point follow below.

Comment 1.1:

“1) Terminology At several points (incl. figure captions) the authors refer to the “omega” parameters as “regulatory strengths”. This is somewhat misleading, no? Wouldn’t it be better to call these parameters “transmissibility” or “susceptibility” or similar?

It seems like it would actually be $(1 - \omega)$ or $1/\omega$ that would properly be called “regulatory strength”.

Response to 1.1:

Thank you for this insightful observation. We fully agree that the original terminology could be inconsistent and potentially confusing. In the **Model** section of the original manuscript, ω was already correctly defined as "effective transmissibility". However, we acknowledge that in other parts of the manuscript, particularly in figure captions and the Discussion section, the terminology was inadvertently mixed with "regulatory strength", which could mislead readers.

Action 1.1:

In the revised manuscript, we have systematically distinguished between these two concepts. This clarification has been propagated through all figure captions and the Discussion to ensure terminological consistency. These changes are highlighted in the revised manuscript, where corrections in the Model, Methods sections, and all figure captions are clearly indicated.

Comment 1.2:

2) The cost function

The authors assume a cost-of-control function that is exponential in the control parameters (the omegas). I get that this has the nice property of taking values in $[0, 1]$, but I think the justification for this particular shape of cost function is on the thin side. (Yes, it resembles a Boltzmann factor, but I’m not sure social dynamics should a priori obey the canonical ensemble or similar!)

Would the authors’ main findings (e.g. the “sweet spot” of intermediate modularity) persist under other cost functions (e.g. linear, quadratic)? Does the exact cost function even matter, or is it sufficient e.g. that it be monotone increasing in both omegas?

Response to 1.2:

We thank you for this insightful comment. We agree that the original reference to a Boltzmann-like function was potentially misleading, as our framework is not grounded in thermodynamic principles.

In our framework, the **regulatory strength** is defined as $(1 - \omega)$. The exponential cost function was introduced to represent the realistic situation in which intervention costs increase nonlinearly as regulation approaches complete suppression: moderate intervention is relatively inexpensive, whereas enforcing near-total blockage typically requires

disproportionately large effort. The exponential form serves merely as a convenient bounded and convex function to represent this empirical observation, rather than implying any thermodynamic mechanism.

To address your concern, we tested alternative cost functions, including a **linear form** ($\mathcal{F} = \frac{(1-\omega_{\text{intra}})+(1-\omega_{\text{inter}})}{2}$) and a **quadratic form** ($\mathcal{F} = \frac{(1-\omega_{\text{intra}})^2+(1-\omega_{\text{inter}})^2}{2}$). As shown in **Figure R1 (and now included in the revised SI Fig. S4)**, all three formulations yield a consistent non-monotonic profile peaking at $\mu \approx 0.5$, confirming that the optimal intervention domain is a **robust topological feature** independent of the cost functional form.

Among the tested options, the exponential form provides a convenient bounded representation and emphasizes the rapidly increasing cost near complete suppression (Panel b of Fig. R1), which justified its use in the manuscript.

Figure R1. **Sensitivity analysis of the optimal cost $\mathcal{F}(\omega_0)$** under three different forms of cost functions, Linear (blue triangles), Quadratic (orange squares), and Exponential (green circles), as a function of the mixing parameter μ (by TL approximation on ER-ER-ER network). **(a)** Optimal cost: the absolute magnitudes differ across functional forms, but all three curves exhibit a consistent non-monotonic profile with a single maximum at intermediate μ . **(b)** Normalized optimal cost: after rescaling each curve to its own maximum value, the three profiles collapse onto a similar shape, confirming that the peak region consistently occurs at intermediate μ . While the detailed curvature reflects the different sensitivities of the cost functions to the regulatory parameters, the qualitative conclusion remains unchanged: the existence of an optimal intervention regime at intermediate modularity is a robust structural feature and does not depend on the specific form of the cost function. Parameters: $\theta = 0.1, \rho_0 = 0.17, z = 20$.

Action 1.2:

Following your constructive feedback, we have completely removed the thermodynamic/Boltzmann analogy from the manuscript. In its place, we have added a concrete socio-economic justification for the exponential shape and explicitly stated that our conclusions are robust to alternative monotonic cost functions. To thoroughly validate this, we have also updated Supplementary Fig. S4 (shown above as Fig. R1) to explicitly include side-by-side results for linear, quadratic, and exponential cost functions.

Specifically, we have updated the **Optimal intervention domain** section (Page 8, Paragraph 2) as follows:

"The exponential form is specifically chosen to capture a realistic socio-economic scenario in which intervention costs escalate nonlinearly as regulation approaches complete suppression. Moderate interventions are relatively inexpensive, whereas enforcing a near-total blockage of transmission requires disproportionately greater effort and resources. Although this form provides a mathematically convenient, bounded, and convex representation of regulatory burden, our main findings do not rely on its specific mathematical shape. Robustness tests using alternative cost definitions, such as linear and quadratic forms (see Supplementary Fig. S4), yield a consistent non-monotonic profile for the optimal cost. This confirms that the observed "sweet spot" at intermediate modularity is a robust topological feature, independent of the exact cost function, provided that the cost increases strictly with regulatory strength."

We believe these revisions directly address your concern and considerably strengthen the theoretical robustness of our paper.

Comment 1.3:

3) The assumption of two modules I understand that the authors focus on networks of two modules for simplicity, as there is already a lot going on with their modeling (both topology and dynamics). But of course, real social networks that are modular likely have MORE than two communities.

I think a small simulation in a system with more modules (e.g. a stochastic block model) would provide assurance that the authors' findings are universal. Even if there are some simplifying assumptions (e.g., omega-inter is the same between ALL modules, similar for omega-intra). Such a simulation might also reveal even more rich behavior than the authors' have found with just two modules.

Response to 1.3:

We thank you for this valuable suggestion. We fully agree that extending the framework beyond two modules is important for demonstrating the generality of our findings.

Action 1.3:

We have generalized the theoretical framework (**Method Section B**) from the case of $n = 2$ modules to an arbitrary number of modules n . The corresponding Tree-Like (TL) equations describe the evolution of the cavity probability y_l^i in the l -th community,

$$y_{l+1}^i = \rho_0^l + \frac{(1 - \rho_0^l)}{z} \sum_{\{\mathbf{k}\}} \{ P(\mathbf{k}) \times \sum_{i=1}^n [k_i \sum_{m_i=0}^{k_i-1} \mathcal{B}(m_i, k_i - 1, y_l^i) \times \sum_{\{\mathbf{m}\}\{m_i\}} \prod_{j \neq i} \mathcal{B}(m_j, k_j, y_l^j) \times \mathcal{R}(\mathbf{m}, \omega, \theta, \mathbf{k} - 1 \times \hat{e}_l)] \},$$

leading to the steady-state fraction of adopted nodes,

$$\rho_\infty^l = \rho_0^l + (1 - \rho_0^l) \sum_{\mathbf{k}} P(\mathbf{k}) \sum_{\{\mathbf{m}\}} \prod_j \mathcal{B}(m_j, k_j, y_\infty^j) \times \mathcal{R}(\mathbf{m}, \omega, \theta, \mathbf{k}).$$

Where, ρ_0^l denotes the initial adoption ratio, $P(\mathbf{k})$ represents the joint probability density of degrees, \mathcal{B} describes the binomial probability, and the threshold rule \mathcal{R} integrates the generalized intervention function.

Additional simulations on a three-module stochastic block network are presented in the Supplementary Information. To further validate this general formalism, we derived the explicit TL equations and simulated for a system with three interacting modules, a case that was not explicitly illustrated in the original manuscript. Applying the generalized n -module framework to a 3-module network (denoted as A, B, C), the evolution equation for the cavity probability $y^{(A)}$ in module A takes the following form,

$$\begin{aligned} y_{l+1}^{(A)} = & \rho_0^{(A)} + (1 - \rho_0^{(A)}) \sum_{k_{AA}, k_{AB}, k_{AC}} \frac{k_{\text{tot}}}{z} P^{(A)}(\mathbf{k}) \\ & \times \left\{ \frac{k_{AA}}{k_{\text{tot}}} \sum_{\mathbf{m}} \mathcal{B}(m_{AA}, k_{AA} - 1, y_l^{(A)}) \mathcal{B}(m_{AB}, k_{AB}, y_l^{(B)}) \mathcal{B}(m_{AC}, k_{AC}, y_l^{(C)}) \mathcal{R}(\mathbf{m}, k_{\text{tot}} - 1) \right. \\ & + \frac{k_{AB}}{k_{\text{tot}}} \sum_{\mathbf{m}} \mathcal{B}(m_{AA}, k_{AA}, y_l^{(A)}) \mathcal{B}(m_{AB}, k_{AB} - 1, y_l^{(B)}) \mathcal{B}(m_{AC}, k_{AC}, y_l^{(C)}) \mathcal{R}(\mathbf{m}, k_{\text{tot}} - 1) \\ & \left. + \frac{k_{AC}}{k_{\text{tot}}} \sum_{\mathbf{m}} \mathcal{B}(m_{AA}, k_{AA}, y_l^{(A)}) \mathcal{B}(m_{AB}, k_{AB}, y_l^{(B)}) \mathcal{B}(m_{AC}, k_{AC} - 1, y_l^{(C)}) \mathcal{R}(\mathbf{m}, k_{\text{tot}} - 1) \right\} \end{aligned}$$

where $k_{\text{tot}} = k_{AA} + k_{AB} + k_{AC}$ is the total degree of a node in module A . After iterating to the fixed point \mathbf{y}_∞ , the final density of adopted nodes in module A is computed as,

$$\begin{aligned} \rho_\infty^{(A)} = & \rho_0^{(A)} + (1 - \rho_0^{(A)}) \sum_{k_{AA}, k_{AB}, k_{AC}} P^{(A)}(k_{AA}, k_{AB}, k_{AC}) \\ & \times \sum_{m_{AA}, m_{AB}, m_{AC}} \prod_{j \in \{A, B, C\}} \mathcal{B}(m_{Aj}, k_{Aj}, y_\infty^{(j)}) \mathcal{R}(m_{AA}, m_{AB}, m_{AC}, k_{\text{tot}}) \end{aligned}$$

The densities $\rho_\infty^{(B)}$ and $\rho_\infty^{(C)}$ for modules B and C are computed analogously through the corresponding index permutations.

As shown in **Figure R2 (corresponding to Figure S15 in the revised SI)**, the theoretical predictions from the TL approximation show excellent agreement with Monte Carlo simulations across a range of mixing parameters μ . These results confirm that the community structure-regulation coupling mechanism identified in the two-module case persists in multi-module networks. In particular, the phase diagram is a generic structural property of modular diffusion systems rather than a peculiarity of the two-module setting.

Figure R2. Validation of the community structure--regulation coupling in three-module network. Comparison between theoretical predictions from the TL approximation (a-d, top row) and Monte Carlo simulations (e-h, bottom row) for a 3-module network with mixing parameters $\mu \in \{0.2, 0.4, 0.6, 0.8\}$. We assume a symmetric configuration where the control parameters are uniform (identical ω_{intra} for all modules and identical ω_{inter} between all modules) and the mixing parameter μ is consistent across all inter-module connections. The fixed parameters are $\theta = 0.1$, $\rho_0 = 0.17$, and average degree $z = 20$. Simulations were performed on networks of size $N = 20,000$. The excellent agreement between theory and simulations confirms that the modularity-controllability coupling identified in the two-module case extends to multi-module networks, supporting the robustness of the proposed theoretical framework.

Comment 1.4: (Remarks on code availability)

I have only verified that the code for all simulations advertised seems to be there. I have not run any of the ipython notebook files myself.

I would advise the authors to do the following:

It's a bit unorthodox to store a .zip archive in a github repository. It would be better to structure the code in a directory, and keep only source (text) files, deleting any other files such as data that can be generated by the user.

Similarly, this repository seems quite large (~100 MB uncompressed). Much of that seems to be the data (as in 1 above), but it's also likely that your notebook files were committed with metadata/outputs (e.g. figures) included). There are also some PDF files (your figures) that, presumably are generated from the code and hence don't need to be stored in the repository.

Response to 1.4:

Thank you for carefully examining the code repository and for these helpful suggestions regarding repository organization.

Action 1.4:

Following the reviewer's advice, we have reorganized and streamlined the repository. Specifically, we have: (1) Unpacked the code into a proper directory structure; (2) Removed large binary data files and redundant PDF outputs; and (3) Cleared output cells from Jupyter notebooks to substantially reduce the repository size.

The updated repository now contains only the essential source files.

Response to Reviewer #2

Comment 2.0:

The paper tackles an important problem: controlling the spread of complex contagion with minimum cost. The work does not provide a concrete, applicable intervention strategy, rather it studies a mathematically tractable model of information diffusion and explores mechanisms that influence the dynamics. While I find the work interesting, I believe that the paper under-delivers what it promises in the introduction and that it could be strengthened by more carefully explaining the mechanisms behind the observed phenomena.

Response to 2.0:

We thank you for this important observation. We fully agree that our work does not aim to provide a concrete, directly deployable intervention strategy for controlling rumor or misinformation spreading. Instead, the goal of this study is to develop a mathematically tractable framework that clarifies how modular network structure interacts with **regulatory parameters** (ω_{intra} , ω_{inter}) to shape diffusion dynamics. In this sense, the work is intended to identify structural mechanisms and theoretical limits of regulation, rather than to prescribe a specific engineering intervention.

We acknowledge that the original introduction may have suggested a stronger level of practical applicability. In the revised manuscript, we have therefore reframed the Introduction to more clearly position our contribution as a theoretical analysis of the community structure-regulation landscape governing complex contagion dynamics.

In addition, we have strengthened the discussion of the underlying mechanisms. Specifically, we now explicitly acknowledge the connection to the seminal work of Nematzadeh et al. (2014) and clarify how our framework extends their model by introducing regulation parameters (see Comment 2.1); we have also refined the conceptual framing by removing references to co-evolution and focusing on static modular structures (Comment 2.2); and replacing ambiguous terms such as "stability" and "resilience" with precise statements about robustness (Comment 2.3); demonstrating that the main findings are insensitive to the specific threshold formulation through additional constant-threshold analysis (Comment 2.4); clarifying the role of the cost function as a general theoretical metric rather than a specific policy model (Comment 2.6); validating the robustness of the results across heterogeneous degree distributions (Comment 2.7).

Comment 2.1:

1) The submitted article is closely related to and builds on the setup and results of Nematzadeh et al., 2014 (Ref [23]), which studies identical spreading dynamics and community structure, uses the same analytical solutions, and is the original paper pointing out the potentially counter-intuitive role of modularity. The current paper adds a control mechanism to this setup, but essentially studies the same phenomenon. I think this must be acknowledged more clearly than it is now and this Ref [23] should be cited more prominently.

Response to 2.1:

We thank you for highlighting the important work of Nematzadeh et al. (2014) and fully agree that it represents a **seminal** contribution to the study of complex contagion in

modular networks. In the revised manuscript, we now cite this reference more prominently and explicitly acknowledge its foundational role in establishing how modular network structure shapes cascade dynamics. While our study adopts a related contagion model and analytical framework, the central question we address is different. Nematzadeh et al. focused on how community structure affects uncontrolled spreading under fixed transmission conditions, whereas our work introduces explicit regulatory parameters for transmission within and between communities. **This substantial extension allows us to move beyond characterizing diffusion on modular networks and to ask how regulation should be configured for a given community structure.** In this sense, our work builds on the foundation established by Nematzadeh et al., while adding **a new intervention-oriented dimension through differential intra- and inter-community regulation and the identification of a cost-efficient optimal controllability domain.** To clarify this relationship, we summarize the main conceptual differences below.

Aspect	Nematzadeh et al. (2014)	Our Work
Research Question	How does modularity affect uncontrolled spreading?	How does modularity interact with regulatory parameters (ω_{intra} , ω_{inter}) to shape diffusion?
Control Parameters	None; transmissibility is fixed	Differential regulation of intra- and inter-community transmission: $\omega_{intra} \neq \omega_{inter}$
Key Output	Phase diagram of spreading vs. modularity	Optimal intervention domain and cost-efficient regulatory configurations
Practical Relevance	Understanding spreading dynamics in modular networks	Providing a theoretical framework for topology-aware regulation in modular systems

Action 2.1:

We have revised the Introduction to more clearly acknowledge Nematzadeh et al. (2014) as a foundational study of complex contagion in modular networks. In the revised manuscript, we added a dedicated paragraph explaining that Nematzadeh et al. showed how modular network structure alone can shape cascade dynamics and generate nontrivial spreading behavior. We then clarify that our work builds directly on this framework by introducing explicit regulatory parameters (ω_{intra} , ω_{inter}) that modulate transmission within and between communities. This substantial extension allows us to move beyond uncontrolled spreading and examine how modularity interacts with regulation, leading to the emergence of an optimal intervention domain in which diffusion can be suppressed at minimal regulatory cost, a feature absent from the uncontrolled setting considered in Ref. [23]. This clarification has been incorporated into the Introduction (Page 3, Paragraph 2) of the revised manuscript, where Ref. [23] is now cited more prominently and the relationship between the foundational framework and our regulation-based extension is stated explicitly. The relevant revised text now reads:

“Recent studies have shown that modular network structure can fundamentally reshape the dynamics of complex contagion. In particular, Nematzadeh et al. [23] demonstrated that community structure can produce nontrivial cascade behavior, revealing that intermediate modularity may maximize the probability of global spreading. While their work provided a foundational framework for understanding uncontrolled diffusion dynamics, how structure interacts with explicit regulatory interventions remains largely unexplored.

To address this gap, we develop a general community structure–regulation coupling framework (COSREF) (Fig. 1). As a substantial extension to the aforementioned modular contagion models, we introduce an additional dimension to the problem by incorporating explicit parameters that modulate transmission. Specifically, our framework links (a) community structure, (b) threshold-based contagion dynamics, and (c) two regulatory parameters, ω_{intra} and ω_{inter} , which govern intra- and inter-community influence, respectively.”

Comment 2.2:

2) The introduction identifies two challenges in controlling complex contagion: heterogeneity of the network structure and co-evolving dynamics and network structure. Heterogeneity is somewhat considered: although the main analysis focuses on Poisson degree distribution, some real networks are considered. However, dynamics co-evolving with network structure are not considered at all, leaving the reader wondering why it is so prominently discussed in the introduction?

Response to 2.2:

We thank you for this valuable observation. Regarding network heterogeneity, our study already examines several network structures beyond Poisson degree distributions, including heterogeneous configurations such as ER–SF–ER and SF–SF–SF networks, as well as empirical social networks. To make this robustness more visible to readers, in the revised manuscript we have **moved the Scale-Free (SF–SF–SF) results from the Supplementary Information to the main text, as shown in Fig. 4**, allowing the robustness of our conclusions under heterogeneous degree distributions to be directly verified.

Regarding co-evolving network dynamics, the reviewer is correct that our current study focuses on diffusion dynamics on static modular networks, rather than networks that evolve together with the spreading process. The reference to co-evolution in the original Introduction was intended to highlight a broader challenge in controlling complex contagion, but we agree that this aspect was not directly addressed in the present model.

Action 2.2:

To avoid potential confusion, we have revised the **Introduction** to remove the previous emphasis on co-evolutionary dynamics and instead focus the motivation on structural heterogeneity and regulation strategy in modular systems. We also briefly note in the **Discussion** section that extending the framework to adaptive or co-evolving networks (e.g., through rewiring or state-dependent link formation) represents an important direction for future research.

Comment 2.3:

3) *In the abstract the authors claim that their work “ provides general principles for achieving stability, efficiency, and resilience in modular systems”. This should be more explicitly discussed in the paper, because right now I do not see how stability and resilience is related to the results.*

Response to 2.3:

We thank you for this important clarification. We acknowledge that the terms “stability” and “resilience” in our original abstract may have been misleading, as they have specific meanings in the complex systems literature.

To clarify, our work does not investigate “stability” or “resilience” in this technical sense. Instead, we focus on identifying **robust structural mechanisms** governing diffusion control in modular networks. In particular, we show that the optimal intervention domain persists across different network topologies, degree distributions, and parameter regimes, and that the theoretical predictions are consistently supported by Monte Carlo simulations.

Action 2.3:

We have revised the abstract by removing the terms “stability” and “resilience” and replacing them with more precise language that better reflects our contribution: the identification of principles for efficient and robust regulation in modular systems.

Comment 2.4:

4) *The dynamics has three parameters: theta governing the threshold and two control knobs, the weight of inter- and intra-connections. Couldn't the number of the parameters be reduced? The parameters appear in the dynamics as $m \cdot w > \theta \cdot k$, hence one could, for example, normalize the control weights and theta to set one of the average weights to 1.*

I am not sure, but couldn't the control weights be absorbed into the degree distribution, i.e., with the substitution $z \rightarrow w \cdot z$? This would reduce the setup to the same setup studied in Ref [23].

Response to 2.4:

We thank the reviewer's insightful comment. We address the two issues raised below.

1. Parameter reduction ($\theta \cdot k$): We have indeed examined an alternative formulation in which the activation threshold is independent of node degree. As described in the manuscript [Eq. (6) and Fig. S9], we consider a **constant-threshold contagion mechanism**:

$$\mathcal{R}(m, \omega, C) = \begin{cases} 1, & m \cdot \omega > C \\ 0, & \text{otherwise} \end{cases}$$

where C is a fixed activation constant independent of k , which captures processes where adoption depends on an absolute stimulus rather than proportional influence, such as aerosol-based spreading or collective decision cascades. This formulation effectively replaces the proportional threshold $\theta \cdot k$ with a single constant threshold. Our results in Fig. S10 show that for $\mu = 0.2, 0.8$ and $C = 2, 3$, the TL predictions remain very close to those obtained with the proportional-threshold rule. **This indicates that our main**

findings, including the phase structure and intervention regimes, are robust to the specific threshold definition.

2. Absorbing control weights into the degree distribution ($\mathbf{z} \rightarrow \boldsymbol{\omega}\mathbf{z}$): Mathematically, it is possible to absorb the control weights into an effective degree. However, doing so would remove the key conceptual distinction in our framework. In our model, \mathbf{z} represents the fixed network structure, whereas $\boldsymbol{\omega}$ represents **tunable regulatory parameters** associated with a cost function $\mathcal{F}(\boldsymbol{\omega})$. If the control weights were absorbed into the degree distribution, the regulatory parameters would no longer remain independent variables, and the cost-optimization problem would disappear. The model would then reduce to the uncontrolled spreading framework studied in Ref. [23]. Our goal, by contrast, is to determine how optimal regulatory parameters should be chosen for a given network structure, which requires keeping topology and regulation as separate dimensions.

Comment 2.5:

5) The paper focuses on control, therefore, the choice of w is a central issue. This is discussed in section "Optimal controllability domain". The way I understand, there is a region of w_{inter} and w_{intra} (yellow area) where control is possible and the outcome is the same. So one would like to choose an extremal point of this region that minimizes some cost.

A possible cost function is introduced, but its consequences are not really explored. None of the figures show the optimal control strategy. For example, one could include a marker on the phase diagrams of Fig 3 showing the optimal point within the yellow region. What is the mechanism behind the control? Are there scenarios where intra control is favored over inter? Or the other way around?

Response to 2.5:

We thank you for this insightful suggestion. We agree that visualizing the optimal strategy and clarifying the mechanism behind the optimal control configuration improves the interpretation of our results.

Action 2.5:

To address this point, we have added **Figure S16** to the SI (also shown below as **Figure R3** for the reviewer's convenience), which explicitly identifies the **optimal intervention point** ω_0 that minimizes the cost function $\mathcal{F}(\boldsymbol{\omega})$ within the intervention region. This point corresponds to the location where the cost contour first becomes tangent to the boundary of the intervention domain.

Because the cost function depends symmetrically on ω_{intra} and ω_{inter} , its iso-cost contours are straight lines with slope -1 . The optimal solution therefore occurs at the tangency between these contours and the phase boundary, which geometrically determines the minimal-cost intervention strategy. This construction also clarifies the mechanism underlying the optimal configuration, which depends on the network modularity μ :

- 1) **For networks with pronounced community structure ($\mu < 0.5$):** Communities are densely connected internally but only weakly connected to each other. Diffusion is therefore sustained primarily by intra-community reinforcement. In this regime, the

phase boundary is steep, and the optimal point favors stronger intra-community regulation (lower ω_{intra}).

- 2) **For networks with weak community structure ($\mu > 0.5$):** Inter-community connections become more abundant, and diffusion is increasingly driven by cross-community bridges. The phase boundary becomes flatter, and the optimal point shifts toward stronger inter-community regulation (lower ω_{inter}).

Figure R3: Optimal intervention strategies across different community structures. Each panel shows the $(\omega_{\text{intra}}, \omega_{\text{inter}})$ parameter space for a given mixing parameter μ . The background color map indicates the cost function $\mathcal{F}(\omega)$, the light blue shaded region marks the non-diffusion phase, and the solid black line denotes the phase boundary. The parallel colored lines (decreasing in value from bottom-left to top-right) represent iso-cost contours of \mathcal{F} with slope -1 . The dashed gray line denotes the specific contours tangent to the phase boundary, where the red star marks the optimal operating point that **minimize** \mathcal{F} , with its coordinates and corresponding \mathcal{F}^* value indicated. **(a)** $\mu = 0.2$: the steep phase boundary places the optimum at $(\omega_{\text{intra}}, \omega_{\text{inter}}) = (0.00, 0.85)$, favoring strong intra-community control. **(b)** $\mu = 0.4$: the optimum shifts to $(0.23, 0.34)$ as the boundary becomes less steep. **(c)** $\mu = 0.6$: the optimum moves to $(0.38, 0.21)$, reflecting a transition toward inter-community control. **(d)** $\mu = 0.8$: the flatter boundary drives the optimum to $(0.78, 0.10)$, favoring stronger inter-community regulation. This progression demonstrates a topology-driven shift in the optimal strategy, from prioritizing intra-community regulation in strongly modular networks to emphasizing inter-community control as modular structure weakens.

This analysis shows that the optimal intervention strategy emerges from the geometry of the intervention region, which reflects how modular network topology shapes the dominant diffusion pathways. We have incorporated this clarification, together with the identification of the optimal intervention point (Fig. S16), into the revised Optimal intervention domain section of the main text.

Comment 2.6:

6) *The choice of the cost function seems a bit arbitrary. In its current form it only depends on the values of w , but not on how many links need to be controlled. Wouldn't it make more sense to study a cost that is proportional to the intervention size? I.e., $(1 - w_{inter})z_{inter} + (1 - w_{intra})z_{intra}$. Or something similar.*

Response to 2.6:

We thank you for this excellent and constructive suggestion. We agree that, in real-world scenarios, the cost of regulating diffusion may depend not only on the regulation strength but also on the number of affected links. A cost function proportional to the intervention size,

$$(1 - \omega_{inter}) \cdot z_{inter} + (1 - \omega_{intra}) \cdot z_{intra}$$

would therefore represent a plausible alternative formulation. In the present study, we adopted a simplified intensity-based cost function that depends only on the regulatory parameters $(\omega_{intra}, \omega_{inter})$. This choice was made for two main reasons.

First, our goal is to introduce a **general theoretical framework** that separates structural effects (captured by the network topology) from regulatory intensity (captured by the control parameters). In this setting, the cost function serves as a generic measure of intervention effort that can be adapted to different practical contexts. In real applications, policymakers could incorporate link-dependent costs, such as those proportional to the number of moderated connections, without altering the structure of the framework. Second, our main results do **not depend on the specific functional form of the cost function**. As demonstrated in the Fig. R1 (see our response to Reviewer #1, Comment 1.2), the qualitative conclusions, particularly the existence of an optimal intervention regime at network of intermediate community structure, remain unchanged when alternative cost functions (e.g., linear or quadratic forms) are used. This indicates that the optimal intervention domain emerges from the **geometry of the diffusion phase boundary** rather than from a particular cost metric.

Therefore, while link-dependent cost functions such as the one suggested by the reviewer could certainly be incorporated into the framework, the simplified formulation adopted here allows us to isolate the fundamental mechanism linking modular network structure and regulation.

Comment 2.7:

7) *It is a good robustness check to consider thresholds that do not depend on the degree. It is perhaps not surprising that the authors find unchanged results for Poisson distributed networks, since in a Poisson distribution most nodes have similar degrees*

close to the average. I would expect to see a bigger difference for SF networks.

Response to 2.7:

We thank you for this valuable suggestion. We agree that the effect of the threshold formulation could be more pronounced in networks with heterogeneous degree distributions, such as scale-free (SF) networks. To address this point, we have examined several network structures beyond Poisson-distributed random networks, including **ER-SF-ER** (modules are internally Erdős-Rényi networks, with inter-module connections following a power-law distribution); **SF-SF-SF** (both intra- and inter-module connections follow power-law distributions); and different power-law exponents γ for the SF case.

Across these heterogeneous network structures, we observe the same qualitative behavior: the system exhibits the same three diffusion regimes (non-diffusion, localized diffusion, and global diffusion), and the optimal intervention domain persists across modularity values. These results indicate that the main findings are robust even under strong degree heterogeneity. The corresponding analyses are presented in the **Supplementary Information (Figs. S5–S8)**.

Comment 2.8:

8) The paper would benefit from a paragraph explaining the network generation algorithm, including how the SF networks are generated.

Response to 2.8:

We thank you for this valuable suggestion. We agree that a clear description of the network generation algorithm is important for reproducibility.

Action 2.8:

We have added a new paragraph in the Computational Methods subsection describing the Scale-Free network generation algorithm. Specifically, we employ the configuration model with the following steps: (i) generate a target degree sequence from a power-law distribution; (ii) create a "stub list" where each node index appears a number of times equal to its assigned degree; (iii) stubs are randomly paired to form edges, avoiding self-loops and multiple whenever possible. This procedure generates networks with the desired power-law degree distribution while preserving the modular structure required for our analysis.

Comment 2.9: (Remarks on code availability)

I recommend uploading the code to github directly, not as a zip file.

Response to 2.9:

We thank you for this suggestion. Following this recommendation, we have reorganized the code and uploaded it to GitHub as a properly structured repository rather than as a compressed archive.

Response to Reviewer #3

Comment 3.0:

The authors present an interesting study aimed at characterising the interplay between complex contagion processes (i.e., threshold processes), clustering, and intra/inter diffusion parameters. The study is articulated with numerical simulations (in both synthetic and real networks), several mathematical derivations, and an overall clear/fluid narrative.

The results are interesting, but the framing and their relevance in the real world are overstated with several unsubstantiated claims. In this current form I cannot recommend this work for Nat Comm. I think that Physics journals (maybe even Nat Phys after several clarification/changes) would be better suited for these types of theoretical works.

Response to 3.0:

We thank you for the careful evaluation of our manuscript and for recognizing the clarity of the presentation and the rigor of the analysis. We also appreciate the concern that the original manuscript may have overstated the immediate real-world applicability of the results.

In the revised manuscript, we have **substantially refined the framing** to better align the presentation with the theoretical scope of the study. Rather than suggesting direct practical deployment, we now present the work as a theoretical framework for understanding **how community structure interacts with regulatory parameters to shape diffusion regulation in modular systems**. The emphasis is now placed on identifying structural mechanisms and providing quantitative insights into how intervention strategies depend on the underlying network organization.

We believe that this revised framing more clearly highlights the broader **interdisciplinary** relevance of the study at the interface of **statistical physics, network science, and social-system dynamics**.

To address the reviewer's concern, we have implemented several revisions throughout the manuscript:

- 1) **Refined claims:** We explicitly scope the results as theoretical guidance for intervention design and discuss the present limitations of the framework, together with possible future extensions, in the Discussion section (see Comment 3.14).
- 2) **Improved terminology:** We replaced potentially ambiguous terms with more precise descriptions of the model and its implications (see Comments 3.1 and 3.5).
- 3) **Strengthened validation:** We further tested the robustness of the results through additional analyses, including degree-preserving rewiring of empirical networks and improved visualization of the phase behavior (see Comments 3.6 and 3.11).

Comment 3.1:

A good place to start is this paragraph in the discussion, where the authors wrote.

“Our MOCOF results demonstrate that modular organization can both suppress and

amplify diffusion, depending on the controllability regime.

This duality produces discontinuous transitions between local and global adoption states, a hallmark of cooperative contagion dynamics.

Simulations on real-world social networks further confirm that the observed transitions and control regimes remain robust in empirical topologies.

Crucially, the identified controllable domain offers a theoretical basis for optimizing information moderation, enabling effective containment with minimal intervention cost.

The cost function, inspired by the Boltzmann distribution, captures how regulatory effort scales nonlinearly with intervention strength, linking statistical physics principles with social-system regulation.”

The results summarised in the first sentence are indeed interesting from a theoretical standpoint. However, the use of the word "controllability" (here as well across the paper) is a bit misleading. Indeed, what the authors do is to simulate a complex contagion process in topologies with different clustering and different intra/inter transmissibility. I am not sure whether within these settings we can speak about "controllability". That word is typically used in the study of dynamical processes where systems dynamics are affected by active (and somewhat realistic) interventions.

Response to 3.1:

We thank you for this important observation. We agree that the term “controllability” has a specific meaning in control theory (Liu & Barabási, 2016, Cornelius et al., 2013), where it typically refers to the ability to steer a dynamical system through external control inputs. Our use of this term in the original manuscript may therefore have been misleading. In the present study, we do not consider node-level interventions or driver-node control in the classical control-theoretic sense. Rather, we examine how tuning the effective transmission parameters (ω_{intra} , ω_{inter}) changes the macroscopic diffusion outcome of the system. In this sense, our analysis focuses on identifying regions of parameter space in which spreading can be **suppressed or confined**, rather than on designing explicit control signals.

Action 3.1:

To avoid confusion, we have revised the terminology throughout the manuscript. In the Introduction and Discussion sections, we now use more precise expressions such as “macroscopic regulation”, “diffusion containment”, and “regulation regimes” when referring to the management of spreading dynamics. These revisions make clear that the manuscript addresses structural regulation of diffusion processes rather than control in the classical dynamical-systems sense.

Comment 3.2:

The second sentence raises another important issue I have with the work presented. Indeed, the authors speak many times about first-order phase transitions. However, they do not really properly investigate the nature of such transitions and base their judgement on the figures. Are we really sure about the nature of those transitions? Do we really need to classify them in this way without an adequate investigation? As far I can tell these transitions are indeed abrupt, but as the "querelle" 15 years ago about the nature

of phase transitions in explosive percolation process shows, deeper investigations are needed to really characterise these types of transitions. On a similar point, also all the claims about “universality” seems to me overselling some observations in a few topologies. Do we really need to claim universality? Do we have any theoretical support for this claim?

Response to 3.2:

We thank you for this important observation. We agree that rigorously classifying the observed transitions requires a dedicated finite-size scaling analysis and cannot be established from the present results alone. In the original manuscript, our use of the term “first-order phase transition” was based on the abrupt jumps observed in the order parameter in both simulations and theoretical predictions. However, we agree that this evidence is not sufficient for a formal classification, and that a more careful analysis is required.

To avoid overinterpretation, we have revised the terminology throughout the manuscript. Specifically, we now describe these phenomena as “**abrupt transitions**” in the adoption density rather than formally classifying them as first-order transitions. We note that Supplementary Fig. S3 examines finite-size effects over the range $N = 2 \times 10^5$ to 10^6 and shows that the phase boundaries and overall phase structure remain stable across the system sizes considered; however, we do not regard this as sufficient evidence to determine the transition order.

Regarding the use of the term “**universality**”, we agree that this wording was too strong. Our intention was only to convey that the main qualitative features of the model, including the existence of distinct diffusion regimes and an optimal intervention domain, remain consistent across the network topologies studied here, including Poisson, scale-free, and empirical social networks. We have revised the wording accordingly.

Action 3.2:

We have replaced “first-order phase transition” with “abrupt transition” throughout the revised manuscript. We have also removed the term “universality” and replaced it with the more precise wording “robustness across the investigated topologies” where appropriate. In addition, we now state explicitly in the Discussion section that a rigorous characterization of the transition order, including finite-size scaling and critical-exponent analysis, is beyond the scope of the present work and remains an important direction for future research.

Comment 3.3:

Regarding the third sentence, the dynamics studied are based on a simple model which is just a rough approximation of real interactions. Hence, can we really claim to have identified a method to effectively control information spreading? Also, how a method based on this paper exactly would work in practice? This appears to me yet another speculation not really supported by evidence, nor needed.

Response to 3.3:

We thank you for this important observation. We agree that the diffusion dynamics studied here are based on a simplified threshold model and therefore provide only an abstraction

of real social interactions. Our intention is not to propose a directly deployable method for controlling information spreading in real-world systems.

Rather, the aim of this study is to identify the basic mechanisms through which community structure and regulatory parameters jointly shape diffusion outcomes. In this sense, the framework is intended to provide theoretical insight into the structural conditions under which diffusion can be more or less effectively regulated, rather than a practical intervention protocol. To avoid overstating the implications of the model, we have revised the manuscript accordingly.

Action 3.3:

In the revised manuscript, we have moderated statements suggesting direct practical control of information spreading and now frame the results as theoretical insight into how community structure influences the efficiency of regulatory interventions in diffusion processes. We have also added a paragraph in the Discussion section to explicitly outline the limitations of the current model and identifying important ingredients required for real-world applications, including temporal network dynamics, heterogeneous behavioral responses, and adaptive network evolution. These revisions make clear that the present work provides a conceptual framework for understanding diffusion regulation in modular systems rather than a ready-to-deploy control method.

Comment 3.4:

Regarding the final sentence, considering the nature of dynamics and systems under study, why do we care about minimizing a Boltzmann-like distribution?

Response to 3.4:

We thank you for this important point. We agree that referring to the cost function as “Boltzmann-like” may create unnecessary associations with statistical mechanics that are not essential to the present study. In our framework, the exponential form of the cost function was chosen simply as a bounded and convex functional form for the optimization problem. This form captures the idea that intervention costs increase nonlinearly as regulation becomes stronger (i.e., as $\omega \rightarrow 0$). Importantly, our conclusions do not depend on this specific functional form. As shown in the sensitivity analysis presented in the Fig. R1 (see our response to Reviewer #1, Comment 1.2), the main results, including the existence and location of the optimal intervention regime, remain qualitatively unchanged when alternative cost functions (linear and quadratic) are used.

Action 3.4:

To avoid misunderstandings, we have removed the statements linking the cost function to the Boltzmann distribution and clarified in the revised manuscript that the exponential form is used only as a convenient convex cost function for the optimization analysis (RESULTS: Optimal intervention domain, Page 8). The relevant revised text now reads:

The exponential form is specifically chosen to capture a realistic socio-economic scenario in which intervention costs escalate nonlinearly as regulation approaches complete suppression. Moderate interventions are relatively inexpensive, whereas enforcing a near-total blockage of transmission requires disproportionately greater effort and resources. Although this form provides a mathematically convenient, bounded, and convex

representation of regulatory burden, our main findings do not rely on its specific mathematical shape. Robustness tests using alternative cost definitions, such as linear and quadratic forms (see Supplementary Fig. S4), yield a consistent non-monotonic profile for the optimal cost. This confirms that the observed "sweet spot" at intermediate levels of community structure is a robust topological feature, independent of the exact cost function, provided that the cost increases strictly with regulatory strength.

Besides, we deleted the last sentence of the second paragraph in the Discussion section (original page 9), and the last sentence of the final paragraph in the "II. RESULTS (Optimal intervention domain)" section (original page 7).

Comment 3.5:

The authors decided to use the word modularity as synonymous of clustering. I understand the choice but considering how "charged" the discussion around community detection methods based on modularity metrics is, I would suggest using the word clustering or community structure.

Response to 3.5:

We thank you for this thoughtful suggestion. We agree that the term "modularity" can be conflated with the modularity metric used in community detection, which may lead to unnecessary confusion.

Action 3.5:

To avoid this ambiguity, we have replaced the term "modularity" with "community structure" throughout the manuscript. The parameter μ retains its original definition as the mixing parameter, representing the fraction of inter-module edges in the network and thus controlling the strength of community structure.

Comment 3.6:

Figure 2 is one of the most important figures in the paper. The panels b, c, e, f are poor graphic representations, the simulation points are not visible, the lines mix and it is not clear what they show. They need to be better represented, ticker lines, better viz of the simulation points, larger plots might help. Simulations could be shown as shaded areas with median and CI. Much more work is needed here as this is a key figure of the paper. Since the MF does not really work, can we just move it to the SI to help the narration and the figure?

Response to 3.6:

We thank the you for this constructive feedback. We agree that Figure 2 is a central figure of the manuscript and its presentation should clearly convey the comparison between theory and simulations.

Action 3.6:

In the revised manuscript, Figure 2 has been substantially redesigned. Simulation results are now shown as clearly visible discrete markers with error bars, using larger marker sizes and improved contrast relative to the theoretical curves. We also refined the color scheme

and line styles to reduce overlap between curves and to improve visual separation among different parameter sets. Following the reviewer's suggestion, we have moved the MF approximation to the SI, as it serves mainly as a qualitative reference and is less accurate than the TL approximation. The revised Figure. 2 now focuses on the comparison between TL theory and Monte Carlo simulations, which improves the clarity of the presentation and streamlines the narrative.

Comment 3.7:

Interestingly, in figure 2b we see that for small values of μ (high clustering) $\rho \sim 1$, then increasing μ leads to a suppression and then increasing even more leads to another global spreading. So, we are in front of highly non-monotonic dynamics. This is a very interesting result that the authors comment, but I feel a bit more discussion about what might be responsible for this would be extremely beneficial.

Response to 3.7:

We thank you for highlighting this important non-monotonic behavior. A related phenomenon was previously identified in the context of uncontrolled complex contagion on modular networks (Nematzadeh et al., 2014). In our framework, we show that this mechanism persists when differential regulatory parameters are introduced. The underlying mechanism arises from the competition between local reinforcement and global connectivity. In networks with fixed average degree, increasing μ effectively replaces intra-community links with inter-community links. For small μ , dense intra-community connections provide strong local reinforcement, enabling complex contagion to spread widely; At intermediate μ , this local reinforcement is weakened, while cross-community connectivity is still insufficient to sustain global spreading, leading to suppression of diffusion; For large μ , inter-community links become sufficiently abundant to support spreading through cross-community pathways, restoring global diffusion. This mechanism explains the strongly non-monotonic dependence observed in Fig. 2b. It also has direct implications for the intervention strategy: at small μ , regulation of intra-community transmission (ω_{intra}) is more effective, whereas at large μ , regulation of inter-community transmission (ω_{inter}) becomes more important.

Action 3.7:

To clarify this point, we have added a brief mechanistic explanation in the Results section ("Regimes of diffusion and phase transitions", page 7). The revised text describes how increasing μ induces a structural dilution of local reinforcement, which leads to the observed suppression of diffusion at intermediate mixing levels. *"This strongly non-monotonic behavior arises from the competition between local reinforcement and global connectivity. As μ increases, dense intra-community links that sustain complex contagion are progressively replaced by inter-community bridges. This structural dilution initially suppresses diffusion because local reinforcement weakens before cross-community connectivity becomes strong enough to sustain global spreading."*

Comment 3.8:

In figure 2e we see another very interesting result. As $\mu \rightarrow 1$ (hence moving towards unclustered networks) we get a suppression of the spreading despite $w_{\text{intra}}=1$. What

is behind this? Also in this case, more work is needed to describe these very interesting results.

Response to 3.8:

We thank you for raising this important point. First, we clarify the definition of the mixing parameter μ used in our model. The parameter μ represents the fraction of inter-community links in the network. Thus, $\mu = 0$ corresponds to fully separated communities where all links are intra-community, and $\mu \rightarrow 1$ corresponds to a network where nearly all links connect different communities and the community structure effectively disappears. In this high- μ regime, intra-community links become extremely rare. As a result, even when $\omega_{\text{intra}} = 1$, the influence of intra-community transmission is negligible because very few such links remain. The spreading dynamics are therefore governed almost entirely by inter-community transmission. Under these conditions, diffusion can still be suppressed if the inter-community transmissibility ω_{inter} is insufficient to sustain propagation across communities. This explains why spreading may remain limited in Fig. 2e even when $\omega_{\text{intra}} = 1$.

Action 3.8:

To clarify this point, we have added an explicit explanation of the mixing parameter μ in the the Results section (Subsection: Model) in page 4 and emphasized that increasing μ corresponds to progressively replacing intra-community links with inter-community links. The relevant revised text now reads:

Small (large) values of μ correspond to networks with strong (weak) modular structure, whereas $\mu = 0.5$ represents a random-like connectivity pattern; explicitly, $\mu = 0$ corresponds to fully isolated modules (all intra-connections) and $\mu = 1$ corresponds to a bipartite graph with no community structure (all inter-connections).

This clarification helps explain why diffusion can be suppressed at large μ when the spreading process becomes dominated by inter-community transmission.

Comment 3.9:

The section "Optimal controllability domain" is a bit confusing. As mentioned, before it is not clear what "control" means and how is that applied to the system. The authors say that the function $f(w)$ is used to quantify the "socio-economic" costs, but then we are shown something akin to the Boltzmann distribution. It is unclear however why this is a good choice nor where the "socio-economic" costs are considered. Outside the community of stats phys this choice, without proper motivation, will appear rather strange.

Response to 3.9:

We thank you for this important feedback. We agree that the original terminology was confusing. First, as noted in our response to Comment 3.1, our analysis does not address control in the classical control-theoretic sense. Instead, we examine how tuning the parameters influences the macroscopic diffusion outcome. The goal is therefore to identify parameter regions where diffusion can be effectively suppressed through regulatory interventions, rather than designing explicit control signals.

Second, regarding the cost function, as we clarified in our response to Comment 3.4, the exponential form serves as a **bounded, convex cost function** commonly employed in optimal control problems (Lenhart & Workman, 2007). It captures the **principle of diminishing returns**: the marginal cost of enforcing stricter control increases as the system approaches complete suppression. This is a standard modeling choice in optimization, independent of statistical mechanics or thermodynamic interpretations.

Under this revised framing, the “Optimal Intervention Domain” refers to the region of parameter space in which diffusion can be suppressed at relatively low intervention cost. This interpretation makes the optimization problem and its scope more explicit.

Action 3.9:

To clarify this section, we have revised both the terminology and the presentation in the manuscript. In particular, the section previously titled “Optimal Controllability Domain” has been renamed “Optimal Intervention Domain” to better reflect its meaning. We have also revised the accompanying text to remove references to Boltzmann-type interpretations and to clarify that the cost function is used only as a convenient proxy for intervention effort. The relevant revised text (on page 8, last paragraph) now reads:

“The exponential form is specifically chosen to capture a realistic socio-economic scenario in which intervention costs escalate nonlinearly as regulation approaches complete suppression, thus $1 - \omega \rightarrow 1$. Moderate interventions are relatively inexpensive, whereas enforcing a near-total blockage of transmission requires disproportionately greater effort and resources. Although this form provides a mathematically convenient, bounded, and convex representation of regulatory burden, our main findings do not rely on its specific mathematical shape. Robustness tests using alternative cost definitions, such as linear and quadratic forms (see Supplementary Fig. S4), yield a consistent non-monotonic profile for the optimal cost. This confirms that the observed “sweet spot” at intermediate levels of community structure is a robust topological feature, independent of the exact cost function, provided that the cost increases strictly with regulatory strength.”

Comment 3.10:

The authors speak about “modularity-controllability coupling mechanisms”. However, it is still not particularly clear which mechanism they are referring to as dynamics are not linear and clearly dependent on the phase space. So, what they mean by this?

Response to 3.10:

We thank you for raising this point. We agree that the phrase “modularity–controllability coupling mechanism” could be misleading in the original manuscript.

In our work, the term was intended to describe how network community structure influences the effectiveness of regulatory parameters (ω_{intra} , ω_{inter}) in shaping the diffusion outcome. Specifically, the mixing parameter μ , which characterizes the strength of community structure, alters the relative importance of intra-community reinforcement and inter-community transmission. As a result, the effectiveness of regulating ω_{intra} versus ω_{inter} depends strongly on the underlying community structure. In this sense, the “coupling” refers to the interdependence between network topology and the impact of regulatory parameters on diffusion dynamics, rather than to a distinct dynamical mechanism in the

classical sense.

Action 3.10:

To remove this ambiguity, we have revised the manuscript to replace the phrase “coupling mechanism” with clearer language emphasizing the topology-dependent sensitivity of diffusion outcomes to the regulatory parameters. In particular, we now explain that the analysis characterizes how community structure reshapes the effect of regulatory interventions, rather than proposing a new dynamical mechanism. Corresponding revisions have been made in the Results and Discussion sections:

1. In the "Robustness and validation across real-world networks" section (Page 9, Paragraph 2), we revised the paragraph as follows: "*To examine how community structure and intervention processes reshape the sensitivity of diffusion to regulatory parameters, we systematically evaluate the robustness...*" "*A similar phase behavior is observed in random-regular (RR-RR-RR) networks (Fig. S8), confirming that the observed transitions arise from the intrinsic interdependence between community structure and regulation rather than from artifacts of specific connectivity patterns.*"

2. In the "Discussions" section (Page 12, Last Paragraph): We revised the macroscopic outlook as follows: "*Beyond information diffusion, this interdependence between community structure and the effectiveness of regulatory parameters may also be relevant to a broader class of spreading and cascading phenomena.*"

Comment 3.11:

In the real networks we have only one value of μ for each graph. The authors could try to reshuffle the networks to increase/decrease μ to study the impact of this variable starting from a real topologies. It would be also good to see even just in the SI a similar figure to figure 4 for the network's models.

Response to 3.11:

We thank you for this valuable suggestion.

Action 3.11:

Following the reviewer's recommendation, we performed a rewiring experiment on the empirical networks to systematically vary the mixing parameter μ while preserving the degree distribution of all nodes. This procedure allows us to investigate the impact of community mixing starting from the original real-network topologies.

The detailed phase diagrams and cross-sections are presented in the revised SI as Figures S12–S14 (corresponding to Figures R4–R6 below) for the Friendster, YouTube, and Orkut networks, respectively. Across the full range of reshuffled μ values, the three canonical diffusion regimes—non-diffusion, localized, and global—persist, and the invention domain observed in the synthetic network models is faithfully reproduced.

To highlight this result, we have also added a new discussion paragraph in the main text (Results section: "Robustness and validation across real-world networks", page 10, paragraph 3), which summarizes the reshuffling results and explains that the community structure–regulation coupling is not an artifact of idealized network models but emerges

robustly from the interplay between community structure and regulatory parameters in real social networks.

“A key limitation of empirical networks is that the mixing parameter μ is fixed by the observed topology, preventing direct exploration of its influence on diffusion regimes. To overcome this, we apply a degree-preserving edge reshuffling procedure to each of the three real-world networks (Friendster, YouTube, and Orkut), systematically increasing or decreasing μ while retaining the original degree distribution. The resulting phase diagrams and cross-sections (Figs.~S12–S14) confirm that the three canonical diffusion regimes—non-diffusion, localized, and global—persist across the full range of reshuffled μ values, and that the intervention domain observed in synthetic models is faithfully reproduced on real topologies. These reshuffling experiments thus provide direct empirical evidence that the community structure–regulation coupling is not an artifact of idealized network models but emerges robustly from the interplay between community structure and regulatory parameters in real social networks.”

Figure R4: Diffusion dynamics on the reshuffled Friendster social network with tunable community structure. To investigate the impact of community structure, we employ a rewiring algorithm that preserves the degree distribution of the original network, while systematically varying the mixing parameter μ . This procedure randomizes the specific connections but maintains the statistical properties of the degree distribution. The figure displays phase diagrams (left columns) and cross-sections (right columns) of the final adoption density ρ_∞ for various μ . The cross-sections show ρ_∞ as a function of the intra-community transmissibility ω_{intra} for fixed inter-community transmissibility $\omega_{\text{inter}}=0.1$ (pink), 0.3 (orange), and 0.5 (green). The results show that the diffusion regimes and intervention patterns observed in the synthetic network models remain consistent across different levels of community mixing in the reshuffled empirical network.

Figure R5: Diffusion dynamics on the reshuffled YouTube social network with tunable community structure. The experimental setup, degree-preserving rewiring procedure, and plotting conventions are identical to those used for the Friendster network (Fig. S12). The results show that the diffusion regimes and intervention patterns remain consistent across different levels of community mixing in this empirical network.

Figure R6: Diffusion dynamics on the reshuffled Orkut social network with tunable community structure. The experimental setup, degree-preserving rewiring procedure, and plotting conventions are identical to those used for the Friendster network (Fig. S12). The results show that the diffusion regimes and intervention patterns remain consistent across different levels of community mixing in the Orkut network.

Comment 3.12:

Why are the main results obtained using Poissonian degree distributions? They are not really a good description of real (aggregated) interaction patterns.

Response to 3.12:

We thank you for this important observation. We agree that Poisson distributions do not capture a realistic description of many empirical interaction networks, which may exhibit heavy-tailed degree distributions. Our study was designed to proceed in stages. We first used Erdős–Rényi (Poisson degree) networks as a simple baseline to isolate the core diffusion mechanisms in modular networks. We then examined more heterogeneous network structures, including scale-free networks, and finally validated the results on large empirical social networks. In the original manuscript, the scale-free results were presented in the Supplementary Information because the qualitative behavior of the system was found to be similar across the different degree distributions. However, we agree that showing these results more prominently improves the clarity of the presentation.

Action 3.12:

In the revised manuscript, we moved the scale-free network results (power-law exponent $\gamma = 3.0$) from the Supplementary Information to the main text, where they are now presented alongside the Poisson network results. This revision makes it clear that the main conclusions remain consistent across networks with both homogeneous and heterogeneous degree distributions. The manuscript therefore now presents results across three levels of network complexity: Erdős–Rényi networks (Fig. 3), scale-free networks (Fig. 4, relocated from the SI), and empirical social networks (Fig. 5).

Comment 3.13:

In the methods the authors wrote:

“Since sparse inter-community connections can obscure phase-transition structures and weaken validation, we carefully selected pairs of communities exhibiting substantial inter-community connectivity.”

This appears very close to handpicking to me, which is very peculiar considering the claims of “universality” the authors make also in these networks. Could we see what happens in the randomly selected pairs of other communities?

Response to 3.13:

We thank you for raising this important concern. We agree that the selection procedure should be clearly explained to avoid the impression of biased sampling. In empirical social networks, many pairs of detected communities are connected by very few or no inter-community edges. For such pairs, the spreading dynamics between communities become effectively decoupled, and the parameter ω_{inter} has little or no influence on the diffusion outcome. In these cases, the system behaves as two nearly independent subsystems, making it impossible to meaningfully study the role of inter-community transmission. For this reason, our analysis focuses on community pairs with sufficient inter-community connectivity to ensure that both intra- and inter-community transmission channels contribute to the dynamics. This criterion ensures that the model can capture the interaction between local reinforcement and cross-community spreading that the study aims to investigate.

Action 3.13:

To clarify this point, we revised the Methods section (Subsection: Data) in page 12 to explicitly state the selection criterion. The revised text now explains that community pairs were chosen based on the presence of non-negligible inter-community connectivity, ensuring that the analysis focuses on systems where inter-community diffusion dynamics are observable. *"We focus on community pairs with non-negligible inter-community connectivity. When inter-community links are extremely sparse, the two communities behave as nearly isolated subsystems, and the effect of inter-community transmission becomes negligible. In such cases, the spreading dynamics are dominated by intra-community processes, making it difficult to meaningfully assess the role of inter-community regulation. Our analysis therefore focuses on coupled systems in which both intra- and inter-community transmission channels contribute to the dynamics."*

Comment 3.14:

I found particularly strange the choice to not discuss any limitations. For example, the authors consider static topologies, while we know real dynamics are subject to complex temporal patterns. None of this is considered here. The dynamics are just a simple model which approximate real interactions etc.

Response to 3.14:

We thank you for this important suggestion. We agree that explicitly discussing the limitations of the present framework is essential for placing the results in the appropriate context.

Action 3.14:In the revised manuscript, we have added an Outlook paragraph at the end of the Discussion section (Page 10, last paragraph) that explicitly outlines the main limitations of the present study and directions for future work. Specifically, we now discuss the following points. First, static network topology: the present framework assumes a fixed network structure. In real social systems, however, interaction patterns often evolve over time and may themselves respond to the spreading process, for example through adaptive contact patterns or opinion-driven link rewiring. Incorporating such co-evolving network dynamics is an important direction for future research. Second, simplified contagion dynamics: the threshold model used here provides a stylized representation of social reinforcement. Real adoption processes may involve additional factors, including heterogeneous behavioral responses, information quality, and contextual influences. Third, characterization of abrupt transitions: the model exhibits abrupt changes in adoption density in certain parameter regimes. A rigorous characterization of these transitions through finite-size scaling analysis and related methods remains an important topic for future investigation.

Comment 3.15:

In figure Figure S1 (panel C) the MF solutions show a very strange peak (which appears very off the trend). What's that?

Figure R7: *Elimination of numerical artifacts via forward iteration.* The left column shows the macroscopic density ρ_∞ computed using the heuristic `scipy.optimize.root` solver for different values of the mixing parameter μ (a) and initial states x_0 (c). In certain parameter regimes, this solver exhibits numerical instability, producing spurious peaks near the critical threshold (highlighted in yellow). The right column (b, d) demonstrates the same quantities computed using forward iteration of the dynamical equations, which provides stable convergence and removes the artificial features observed in the solver-based results.

Response to 3.15:

We thank you for this careful observation. The peak observed in Fig. S1 (panel C) was indeed anomalous. Upon further investigation, we found that this feature was a numerical artifact caused by instability in the nonlinear equation solver used in the original implementation (based on `scipy.optimize.root` with the `excitingmixing` method). In certain parameter regimes, this solver can converge to spurious intermediate solutions, producing the artificial peak noted by the reviewer. To address this issue, we replaced the solver with a forward iterative method, which directly evaluates the fixed-point iteration of the mean-field equations. This approach proved to be numerically stable across the entire parameter range and removes the anomalous behavior, as shown in Figure R7.

In addition to replacing the original solver with a forward-iteration scheme, we have performed a systematic validation of the numerical results across the full parameter spaces corresponding to Figs. S1 and S2. As shown in Figs. R8 and R9, the MF results obtained from the forward-iteration method are consistent with those from the original root solver across the parameter space, except in regions where the root solver exhibits numerical instability and produces spurious peaks. The forward-iteration method remains stable in

these regions and removes these artificial features. Importantly, the TL approximation results presented in the manuscript are computed exclusively using the forward-iteration method and are therefore not affected by this numerical issue. These comparisons confirm that the identified anomaly is purely a solver-induced artifact and does not affect the conclusions of the study.

Action 3.15:

We have updated the numerical procedure accordingly, and the corresponding figures in the Supplementary Information have been regenerated using the forward-iteration method. The revised results show a smooth dependence on the initial condition with a well-defined bistable threshold, and the anomalous peak is no longer present. Importantly, we verified that the other results reported in the manuscript remain unchanged, since the main conclusions rely primarily on the tree-like approximation and Monte Carlo simulations.

Figure R8. **Validation of numerical stability** (parameter space corresponding to Fig. S1). Comparison between the original root solver and the forward-iteration method for the Mean-Field (MF) equations. The two methods yield consistent results across the parameter space, except in regions where the root solver produces spurious peaks due to numerical instability. The forward-iteration method remains stable and eliminates these artifacts.

Figure R9. **Validation of numerical stability** (parameter space corresponding to Fig. S2). As in Fig. R8, the forward-iteration method produces stable results across the parameter space and removes the numerical artifacts observed in the root-solver implementation.

Reference:

- [1]. Cornelius, S. P., Kath, W. L., & Motter, A. E. (2013). Realistic control of network dynamics. *Nature Communications*, 4(1), 1942. <https://doi.org/10.1038/ncomms2939>
- [2]. Lenhart, S., & Workman, J. T. (2007). *Optimal Control Applied to Biological Models*. Chapman and Hall/CRC. <https://doi.org/10.1201/9781420011418>
- [3]. Liu, Y.-Y., & Barabási, A.-L. (2016). Control principles of complex systems. *Reviews of Modern Physics*, 54.494, 88(3), Article 3. <https://doi.org/10.1103/RevModPhys.88.035006>
- [4]. Nematzadeh, A., Ferrara, E., Flammini, A., & Ahn, Y.-Y. (2014). Optimal Network Modularity for Information Diffusion. *Physical Review Letters*, 9.161, 113(8), Article 8. <https://doi.org/10.1103/PhysRevLett.113.088701>

Response to Reviewers

We thank all three reviewers for their careful evaluation of our revised manuscript. Below, we address each reviewer's comments in turn.

Response to Reviewer 1

Reviewer #1 (Remarks to the Author):

The authors have satisfactorily addressed my comments in the first round of review. In particular, I commend them for going above-and-beyond in one case, extending their theoretical framework to more than two modules (I had asked only about a simulation).

I recommend publication.

Response: We sincerely thank you for the positive assessment and for recommending publication.

Reviewer 1 (Remarks on code availability):

The authors followed my advice in restructuring their GitHub repository into a more orthodox format. By my standards, the code is sufficiently "available" that a reader could use it. However, I have not taken the time to run the various notebooks, etc., to ensure they produce the intended output. I evaluate their work instead based on the figures in the manuscript and response to reviewers.

Response: Thank you. We greatly appreciate your suggestion in the first round, which helped us significantly improve the organization of the code base. We have also restructured the GitHub repository following your guidance, which substantially improves code accessibility, clarity, and reproducibility.

Response to Reviewer 2

The authors addressed the points that the other referees and I raised, and these changes clarified the context of the paper and made it more readable. I believe that with the added robustness checks the authors obtain close to the maximum of what can be learned from the setup. Having that said, overall much of the changes went in the direction of explaining the limitations of the work and adjusting the claims of the introduction to the actual results.

Response: We thank you for the careful re-evaluation and for acknowledging that the revisions have clarified the context of the manuscript and improved its readability. We also appreciate the recognition that the added robustness checks strengthen what can be learned from the present theoretical framework.

I appreciate that Nematzadeh et al. (2014) is discussed now more prominently and it is explained that the introduction of the w parameter is the main difference. I maintain that it is arguable how substantial this addition is. The manuscript investigates the same phenomena as Nematzadeh et al. (2014) with an additional parameter (that could be mathematically incorporated into the network parameters) and branding w as a control knob. The introduction still reviews previously proposed more realistic regulation strategies and points out issues of practical implementation to motivate the current work. However, the proposed control mechanism is purely theoretical without any discussion of practical implementation.

I do believe that there is much value to theoretical models. In my opinion, however, the results of the manuscript are incremental and lack the level of significant new understanding that the authors claim.

Response: We sincerely thank you for the careful re-evaluation and for the constructive comments across both rounds of review. We appreciate your concerns regarding the positioning of our contribution relative to Nematzadeh et al. (2014), as well as the need to clarify the practical relevance of the proposed framework.

We agree that our work builds on the modular threshold contagion framework and do not claim a fundamentally new spreading mechanism. Instead, our contribution is to reformulate the problem from uncontrolled diffusion to regulated diffusion, by introducing independent intra- and inter-community transmissibility parameters (ω_{intra} , ω_{inter}). This enables us to address a distinct question: how intervention effort should be allocated across different transmission channels for a given community structure. This intervention perspective, and the resulting identification of an optimal intervention domain, is not present in the uncontrolled setting.

We also fully acknowledge the reviewer's concern regarding practical implementation. In response, and following the editor's guidance, we have **extended the Discussion section** to explicitly connect the abstract parameter ω to concrete classes of real-world interventions, while carefully emphasizing the theoretical scope of the present work. The added paragraph (Discussion) now explains how ω_{intra} and ω_{inter} may correspond, for example, to intra-community moderation versus cross-community exposure control in online systems, or to local contact reduction versus mobility restrictions in epidemiological

contexts. At the same time, we explicitly state that the framework does not provide a directly deployable strategy, and that realistic implementation would require temporal dynamics, behavioral heterogeneity, and system-specific constraints.

We are grateful for the reviewer's critical perspective, which has helped us refine both the positioning and clarity of the manuscript.

Action:

Text added to the Discussion section (Page 10): “While our framework introduces ω as a generalized mathematical representation of reduced transmissibility, this parameter can be related to practical interventions in several contexts. In online information systems, decreasing ω_{inter} may correspond to reducing cross-community exposure, for example through downranking, friction, or limits on recommendation pathways that connect otherwise weakly linked groups. Decreasing ω_{intra} may instead represent moderation or exposure reduction within closely connected communities. In epidemic or mobility-driven spreading processes, the same distinction may correspond to local contact reduction within communities versus restrictions on movement between geographic or demographic groups. These examples should not be interpreted as direct prescriptions, since real implementation would require temporal data, behavioral heterogeneity, platform-specific constraints, and ethical considerations. Rather, the framework provides a theoretical way to compare how intervention effort might be allocated between within-community and between-community transmission channels under different community structures.”

Response to Reviewer 3

Reviewer 3 (Remarks to the Author):

The authors did a very good job in the revision. All the points raised in the first round have been addressed. I recommend the article for publication.

Response: We sincerely thank you for the positive evaluation and for recommending publication.